# Light-Activating PROTACs in Cancer: Chemical Design, Challenges, and Applications

Arvind Negi [1,*] , Kavindra Kumar Kesari [1,2,*] and Anne Sophie Voisin-Chiret [3,*]

1 Department of Bioproduct and Biosystems, Aalto University, 02150 Espoo, Finland
2 Department of Applied Physics, School of Science, Aalto University, 02150 Espoo, Finland
3 CERMN (Centre d'Etudes et de Recherche sur le Médicament de Normandie), Normandie University UNICAEN, 14000 Caen, France
* Correspondence: arvind.negi@aalto.fi or arvindnegi2301@gmail.com (A.N.); kesari.kavindra@aalto.fi (K.K.K.); anne-sophie.voisin@unicaen.fr (A.S.V.-C.)

**Abstract:** Nonselective cell damage remains a significant limitation of radiation therapies in cancer. Decades of successful integration of radiation therapies with other medicinal chemistry strategies significantly improved therapeutic benefits in cancer. Advancing in such technologies also led to the development of specific photopharmcology-based approaches that improved the cancer cell selectivity and provided researchers with spatiotemporal control over the degradation of highly expressed proteins in cancer (*pro*teolysis *ta*rgeting *c*himeras, *PROTACs*) using a monochrome wavelength light source. Two specific strategies that have achieved notable successes are photocage and photoswitchable PROTACs. Photocaged PROTACs require a photolabile protecting group (PPG) that, when radiated with a specific wavelength of light, irreversibly release PPG and induce protein degradation. Thus far, diethylamino coumarin for estrogen-related receptor α (ERRα), nitropiperony-loxymethyl (BRD4 bromodomain protein), and 4,5-dimethoxy-2-nitrobenzyl for (BRD4 bromodomain protein, as well as BTK kinase protein) were successfully incorporated in photocaged PROTACs. On the other hand, photoswitches of photoswitchable PROTACs act as an actual ON/OFF switch to target specific protein degradation in cancer. The ON/OFF function of photoswitches in PROTACs (as photoswitchable PROTACs) provide spatiotemporal control over protein degradation, and to an extent are correlated with their photoisomeric state (cis/trans-configuration), showcasing an application of the photochemistry concept in precision medicine. This study compiles the photo-switchable PROTACs targeted to bromodomain proteins: BRD 2, 3, and 4; kinases (BCR-ABL fusion protein, ABL); and the immunophilin FKBP12. Photocaging of PROTACs found successes in selective light-controlled degradation of kinase proteins, bromodomain-containing proteins, and estrogen receptors in cancer cells.

**Keywords:** photochemistry; photocaging; photoswitches; PROTACs; azobenzenes



## 1. Introduction

The electromagnetic (EM) spectrum contains a range of EM radiations. EM radiations (EMR) are composed of microwaves, infrared (IR) and ultraviolet light (UV), X-rays, and gamma-rays, as shown in Figure 1. These EMRs are characterized mainly by frequency and wavelength. As EM radiation contains electromagnetic vectors (energy), which can affect the resonating frequency of biomolecules at the molecular level, it led to the development of advanced cellular imaging and diagnostics technologies. Despite the immense development of these EMR-based technologies, their nonselective cellular damage still limits their direct and prolonged use for chronic treatments. However, applying UV–VIS light found many applications, such as fluorescence, phosphorescence, and photoactivation. Among these, photoactivation has found a broad implementation in targeting a number of cellular disorders by modifying the signaling pathways from GPCR proteins, ligand-gated

ion channels, and other vital cellular components. However, because of the limitation of conventional drug design approaches, there is a paradigm shift in the development of light-activating drug-like molecules. These light-activating molecules provide a spatiotemporal handle over a cellular process. For example, recently, protein degradation approaches showed an application of light to target specific proteins that can be widely used in various types of cancer.

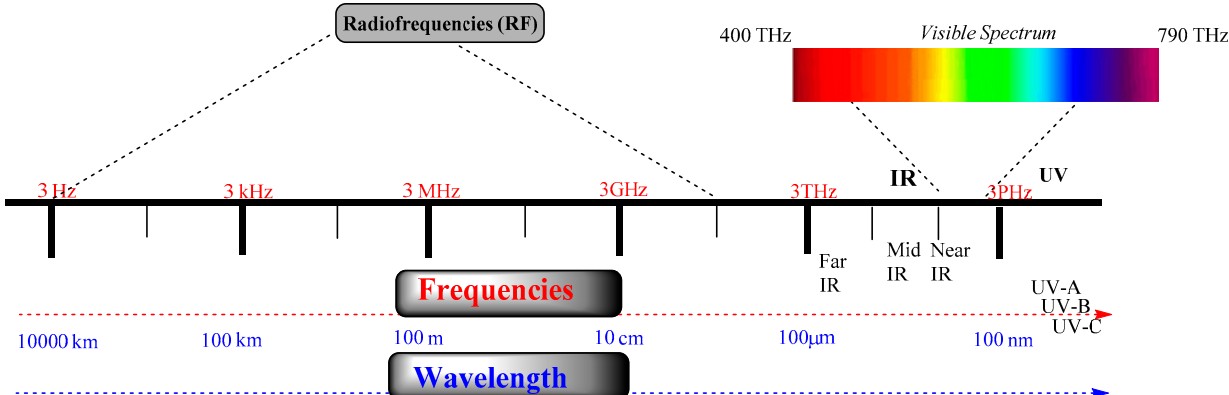

**Figure 1.** The electromagnetic (EM) spectrum range covers frequencies, radiofrequency (3 Hz–300 GHz), microwaves, IR, UV, and gamma-waves.

## 2. Protein Degradation Strategies

Normal cellular functioning requires tight control over the relative rate of biosynthesis and degradation of intracellular proteins. This tight control is continuously monitored by the highly organized intracellular protein signaling systems that commonly comprise multi-partner proteins, communicating in a spatiotemporal way. An alteration in such a sophisticated intracellular cell system leads to catastrophic results, where overexpression of a specific intracellular protein ends up accelerating the disease state advancement. Broadly, strategies devised to eliminate the overexpressed intracellular proteins, where DNA chemical modification leads to protein of interest (POI) knockout at the *gene level* [1], or RNA interference (RNAi) eradicates the POI expression at the mRNA level [2]. However, such approaches take time to complete the POI depletion and encourage researchers to explore the protein-targeting technologies on the basis of event-driven pharmacology. Notable protein-targeting technologies are TRIM-Away [3], HyTs (low-molecular-weight hydrophobic tags) [4], and PROTACs (proteolysis targeting chimeras) [5]. These techniques utilize the protein degradation assembly of the cell to target specific POI. Moreover, these strategies are flawed with irreversible control over protein degradation, insufficient potencies, and low cellular delivery [6]. Among these, PROTAC design strategies found a wide application against several overexpressed proteins in cancer (EGFR [7], CDK9 [8], TRIM24 [9], Bcl-2 family proteins (Bcl-xL [10,11], Bcl-2 [12], Mcl-1 [12,13]), bromodomain [14,15], tyrosine-protein kinase (c-Met [7], ALK [16]), estrogen-related receptor-$\alpha$ [17], and Tau [18]). Furthermore, successful oral PROTAC (ARV-110) targeting androgen receptors entered into a phase-1 clinical study [19]. PROTACs use the intracellular ubiquitination to induce POI degradation. As illustrated in Figure 2, PROTACs are chemically heterobifunctional, made up of three *warheads*, represented as *Part-A* (green), a known drug structure/bioactive molecule that has an affinity towards the *p*rotein *of i*nterest (POI); *Part-B* (*magenta*), the linker region that, in general, works as a connector to connect part-A and part-C; *Part-C* (orange), a chemical structure that has an affinity to E3 ubiquitin ligase (also called as "E3 ligase ligand", for example, cereblon (CRBN) or Von Hippel–Lindau (VHL)). However, subsequent binding of chimeric compounds such as PROTACs (or similar compounds such as SNIPERs) to POI and E3 ligase depends on cooperativity, which means that binding of either warhead to their respective proteins (POI or E3-ligase) influences the binding of the remaining warhead to its partner protein. Therefore, a proportion of bifunctional

compounds (PROTACs, SNIPERs) forms multiple binary complexes either with POI or E3-ligase, even at high concentrations, and such biophysical limitation is also known as a *hook's effect*.

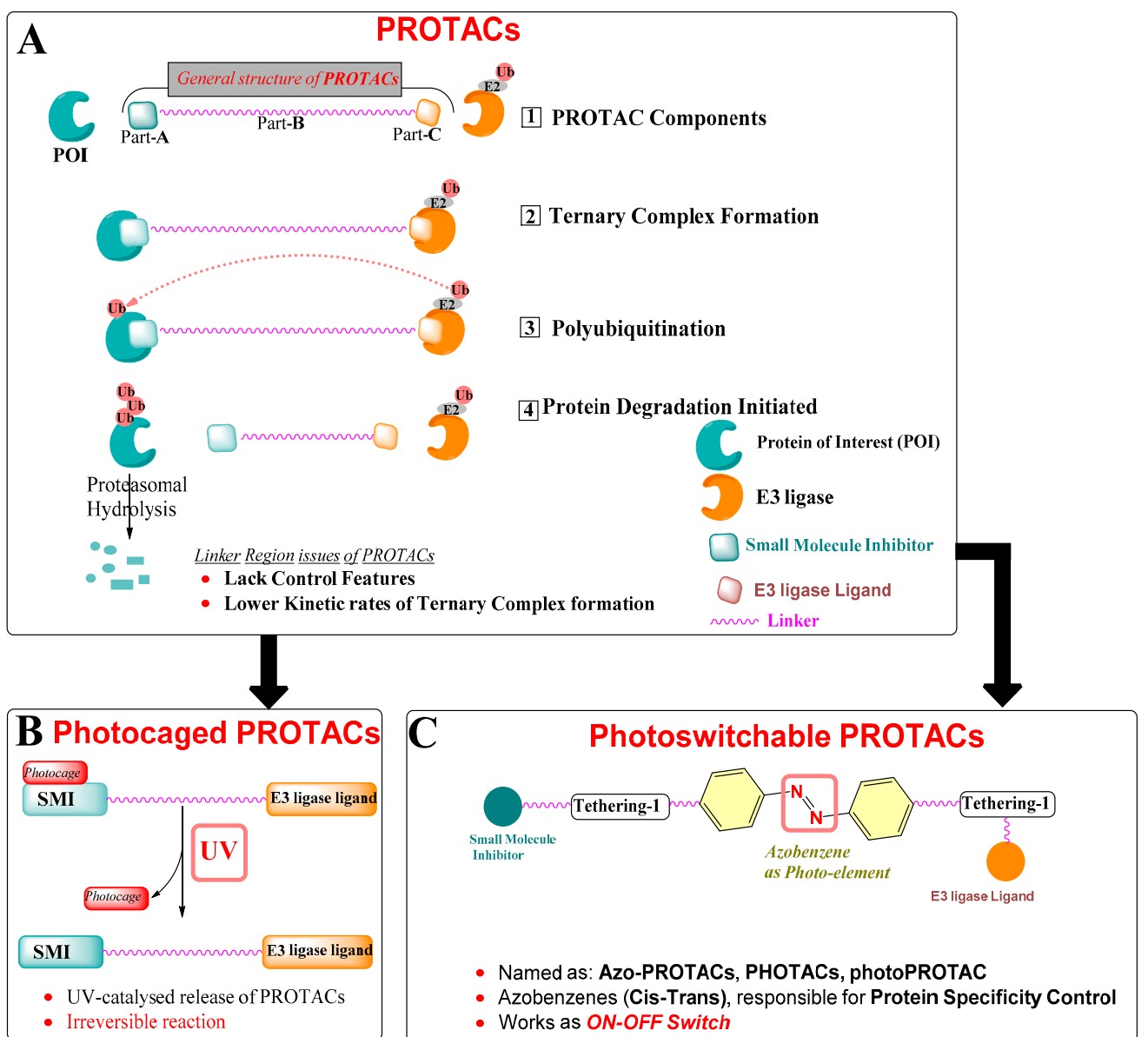

**Figure 2.** (**A**) PROTAC strategy workflow representation in targeted protein degradation. (**B**) Generalized chemical design of Photoswitchable PROTACs. (**C**) Generalized chemical design of Photocaged PROTACs.

The success of PROTACs is hidden in their mechanism of action when compared with the mechanism of action of conventional inhibitors as a conventional inhibitor requires higher binding affinity to engage themselves to the POI, irrespective of PROTACs, which need to hook themselves with POI, allowing E3 ligase in enough proximity to induce the intracellular ubiquitination of POI (illustrated in Figure 2A). The E3 ligase is a ubiquitin protein that recruits an E2-ubiquitin-conjugating enzyme that has ubiquitin units. After identifying the targeted protein, we find the transfer of ubiquitin from the E2 to the lysine amino acids exposed to the surface of the targeted protein that eventually enables POI degradation. After reaching the intracellular site, PROTAC binds to POI through its POI-recognized warhead, while the warhead on the other side of PROTAC recognizes

an E3 ligase protein, ultimately leading to a ternary complex (POI–PROTAC–E3 ligase) formation and facilitating polyubiquitination of the POI, as illustrated in Figure 2A. This polyubiquitin process is an important process that involves transfer of ubiquitin units of the lysine amino acid residues over the POI surface, marking it for its hydrolysis. Therefore, the presence of lysine amino acids over the POI surface plays a decisive role in extending its prolytic hydrolysis.

PROTACs have two salient features in comparison to conventional small-molecule inhibitors (SMI): (a) high potency (sub-micromolar-to-nanomolar) for POI is not a necessary requirement for PROTAC to bind with POI, and therefore incidences of dose-dependent toxicity of PROTACs are quite low and would otherwise also improve their overall therapeutic indexing, (b) PROTACs are unlikely to have the emergence of resistance because of their ability to degrade the POI than inhibit the POI. Therefore, downregulation of POI by any point mutation is less likely possible, and no continuous dosing is required as it is reported that prolonged exposure to POI with inhibitors utilizing occupancy-driven pharmacology alters the utilization of downstream intracellular signaling. Although these advantages show the superiority of PROTACs over SMI-based targeting, they also have their own issues and share design flaws like any medium-to-large-sized heterocyclic structure. As an example, PROTACs have average physicochemical properties because of their molecular obesity (molecular size). Moreover, PROTACs cannot differentiate the normal cells from the disease state cells, and therefore this could cause systemic toxicity and a number of undesired side effects. An example is an excellent regression of castrate-resistant prostate cancer in a mouse model, which was achieved by ARV-771 (a potent BET protein PROTAC), along with skin deterioration at the injection site and systemic cytotoxicity being observed. This exemplifies a requirement for more features incorporated into PROTAC strategies and helps to improve their protein specificity and cell selectivity.

Therefore, photochemical biology tools were explored. On the basis of the incorporation of photoelement type, two strategies were exploited for PROTACs: (a) photoswitches that help in the reversible conformation of PROTACs, where one isomer is biologically active while the other is inactive [20], as shown in Figure 2B, and (b) a photocleavable protecting group (PPG) (caging or photocaging) [21], as displayed in Figure 2C.

### 3. Intracellular Delivery and Controlling Activation/Deactivation PROTACs

The systematic toxicity of PROTAC is one of the primary concerns of the strategy. To achieve cellular selectivity and targeted protein degradation requires efficient intracellular delivery of PROTACs into the cancer cells and avoiding entry to normal cells. One popular idea was to exploit the overexpressed cancer-specific proteins, which would take benefit of highly expressed cell surface proteins, but lesser-to-none expressed on the normal cell surfaces. In this aspect, the antibody–drug conjugate (ADC) strategies showed high selectivity intracellular delivery of PROTACs into those cancer cells expressing cancer-specific membrane hybrid receptors, for example, insulin receptor/insulin-like growth factor receptors [22,23] and heparin-binding epidermal-growth-factor-like growth factor [24,25]. Molecular obesity because of its large molecular size and peptide backbone hydrolysis towards physiological pH are the two limiting factors for ADC approaches for PROTACs. Therefore, new chemical biology tools that can be coupled with PROTAC strategies were investigated. In this aspect, photocaged PROTACs were developed for targeting POI, as shown in Figure 3.

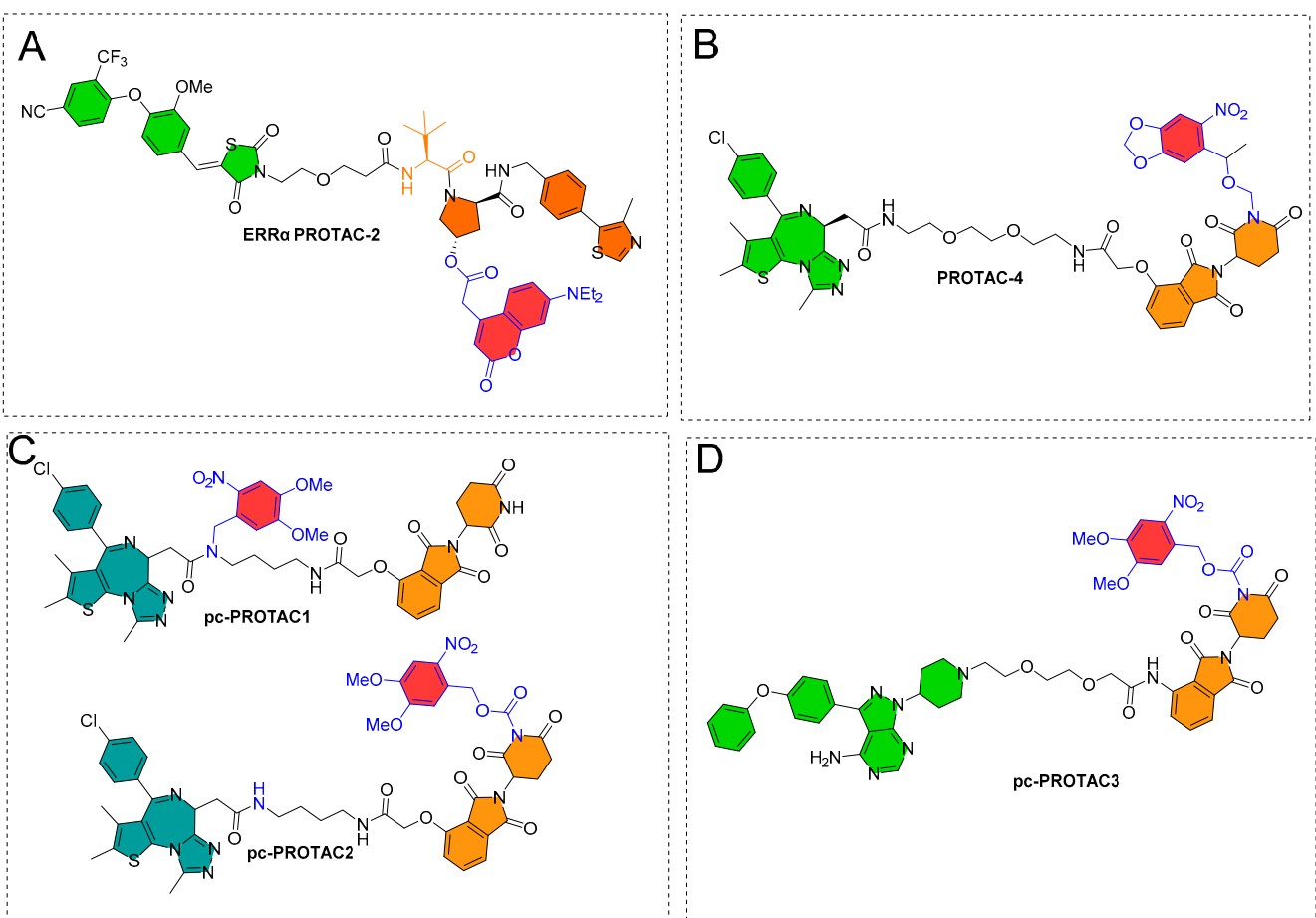

**Figure 3.** Compilation of reported photocaged PROTACs by researchers (**A**). ERRα PROTAC-2 targegting estrogen receptor. (**B**). PROTAC-4 as a bromodomain inhibitor (**C**). pc-PROTAC-1 and pc-PROTAC2 as bromodomain inhibitors. (**D**). pc-PROTAC-3 as a kinase inhibitor.

### 3.1. Photocaging PROTACs of Estrogen Receptor

Gaining precise control over the biological activity of smaller-sized probes has always interested chemical biologists and medicinal chemists. However, a light-controlled higher spatiotemporal resolution has been exploited as a chemical biology tool [26] and in phototherapies [27], where a specific wavelength of light activates the bioactive molecule.

Deiter's research group from the Department of Chemistry, University of Pittsburgh, Pittsburgh, Pennsylvania, USA, developed a coumarin-based photocaged VHL ligand. Initially, the authors investigated an X-ray co-crystal structure (PDB id: 4W9C) to find a tethering point for a photocleavable group onto the VHL ligand [28]. The authors noticed a hydroxyproline moiety buried into the binding cleft that had H-bond interactions with Ser111 and His115, which are critical amino acid residue interactions used to recognize VHL by HIF1-α protein, as shown in Figure 4A [29–31]. Moreover, inverting the hydroxyl group stereochemistry of hydroxyproline moiety abolishes all protein degradability of PROTACs [17,32]. On the basis of these facts, the authors rationalized the suitability of the tethering point for the photolabile group in a way that the tethering of the photolabile group would hinder VHL ligand binding to its VHL E3 ligase and can only be activated until irradiated (as shown in Figure 4B). The approach showcases an example of precise spatiotemporal control over photobiology. The authors used carbonate tethering to substitute the hydroxyl group of VHL with a diethylamino coumarin (DEACM) to form **ERRα PROTAC-2** (as shown in Figure 4C). Moreover, the authors prepared a version of **ERRα PROTAC-2** that does not have a DEACM photolabile group, namely, **ERRα PROTAC-1**, as a control to assist in their biological study and ensure their photocaging PROTAC approach.

Both PROTACs were designed to target an orphan nuclear hormone receptor (estrogen-related receptor $\alpha$ (ERR$\alpha$)) [17], typically overexpressed in malignant cancers [33]. Using HPLC and mass spectrometry, the DEACM caging group cleaved from the **ERR$\alpha$ PROTAC 2** after 3 min of irradiation ($\lambda \leq 405$ nm) and released the acidic functional groups with a pKa < 5 [34].

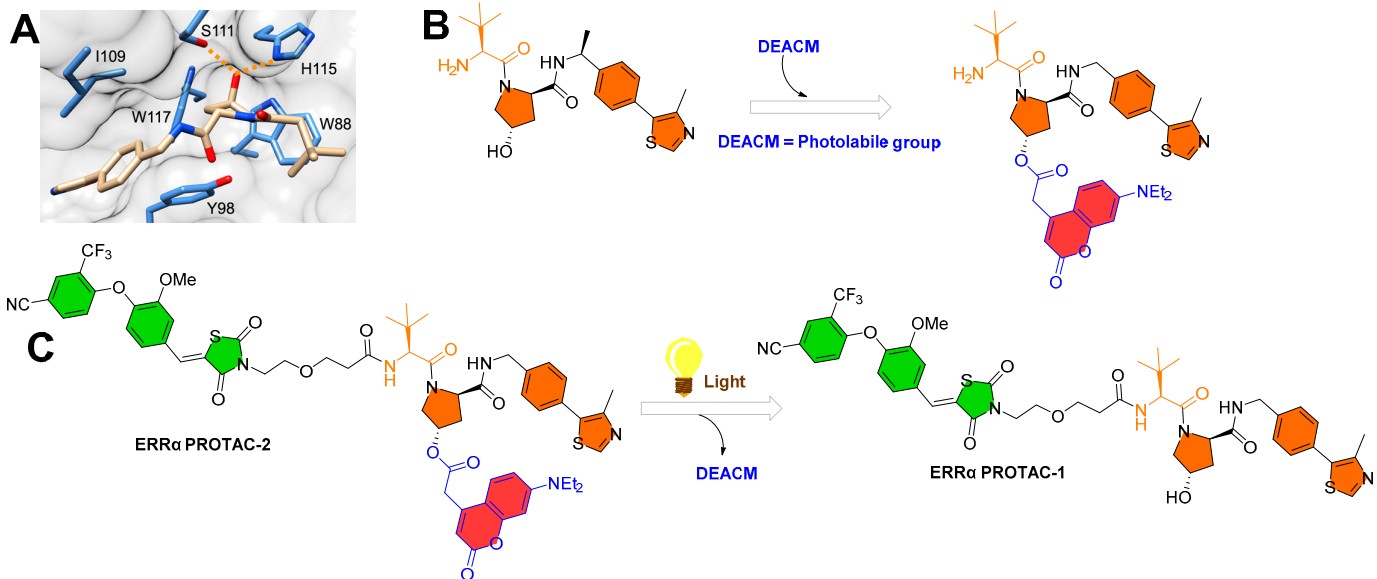

**Figure 4.** (**A**) Cocrystal VHL ligand binding to VHL E3 ligase, showing critical residue involvement in H-bonding interactions (reprinted (adapted) with permission from [35]; Copyright 2020 American Chemical Society). (**B**) Chemical transformation of VHL ligand into coumarin-photocaged VHL ligand. (**C**) Photochemical transformation of **ERR$\alpha$ PROTAC-2** to **ERR$\alpha$ PROTAC-1**.

To understand the photocage PROTAC role of **ERR$\alpha$ PROTAC-2**, MCF-7 cells (breast cancer cell line) were treated with **ERR$\alpha$ PROTAC-1**, **ERR$\alpha$ PROTAC-2,** and DMSO in the absence/presence of UV radiation ($\lambda = 365$ nm, 180 s). After 8 h of incubation, the authors used Western blotting to measure the extent of ERR$\alpha$ cellular levels. As anticipated, **ERR$\alpha$ PROTAC-1** showed a significant ERR$\alpha$ protein reduction at the cellular level compared to the DMSO-based sample (used as control), which agrees with the previously reported literature [17]. Importantly, even a double concentration of **ERR$\alpha$ PROTAC-2** compared to **ERR$\alpha$ PROTAC-1** in the absence of UV light showed no change in ERR$\alpha$ protein cellular levels, confirming that photocaging of the photolabile group prevented the binding conformation towards E3 ligase, exemplifying an idealistic example of photocaged-PROTACs. To understand the mechanism, competitive assays in the presence of either the proteasome inhibitor (**MG132**) or the neddylation inhibitor (**MLN4924**) prevented the degradation of **ERR$\alpha$ PROTAC-2**, suggesting the proteasome- and E3-ligase-mediated degradation ability of **ERR$\alpha$ PROTAC-2**. Furthermore, incubation of MCF-7 cells with the coumarin caging group fragment released during photolysis showed no effects on ERR$\alpha$ levels, demonstrating that the observed degradation activity was highly mediated by the active, non-caged PROTAC generated via decaging.

### 3.2. Photocaging PROTACs of Bromodomain Proteins

To expand the photocaging PROTAC concept, the same authors (Deiter's research group from the Department of Chemistry, University of Pittsburgh, Pittsburgh, Pennsylvania, USA) considered CRBN-based PROTACs. Similar to their chemical design of **ERR$\alpha$ PROTAC-2**, the authors utilized the X-ray structure to find a suitable tethering position for incorporating the photolabile group on the CRBN ligand (as shown in Figure 5A). Through X-ray co-crystallize structure, the authors found imide functionality on the glutarimide ring buried into a hydrophobic pocket and that it had a H-bond interaction with the amide

backbone of His380 (as shown in Figure 5A). A previous CRBN-based PROTAC study [36] reported that substituting even a smaller group such as methyl at imide functionality produces enough steric hindrance that abolishes the binding of overall thalidomide structure with CRBN E3-ligase. These studies led the authors to exploit imide (-NH-) functionality for installing the photolabile group (nitropiperonyloxymethyl (NPOM)) (as shown in Figure 5B). The authors synthesized BRD4 **PROTAC-3**, which has a chemical frame of a reported CRBN-based PROTAC (Bradner's research group from the Dana-Farber Cancer Institute, Harvard Medical School, Boston, USA, reported originally as *compound 3*, with BRD4 $IC_{50}$ = 37 nM by using luminescent-proximity-based AlphaScreen assay [37]). The authors installed the photolabile group (NPOM) onto the imide of glutarimide of BRD4 **PROTAC-3** to obtain the BRD4 **PROTAC-4**. Choosing of the NPOM group was based on its stability in aqueous conditions, as higher acidity of imide nitrogen (pKa = 14) compared to aliphatic amines (pKa = 30–40) was found to be problematic [38,39]. Upon irradiation (λ = 365 nm), the NPOM undergoes photolysis and has already shown its use in previously reported cellular studies [40]. The authors monitored the light activation of BRD4 **PROTAC-4** through HPLC and mass spectrometry, where a non-caged version of PROTAC (**BRD4 PROTAC 3**) and a leaving cage group (nitrosoketone) were observed upon irradiation (at 2 min) (as shown in Figure 5C). The results obtained from cellular experiments with NPOM-caged BRD4 **PROTAC-4** verified its optical control over CRBN-mediated degradation of BRD4, such as (a) a significant degradation of BRD4 in HEK293T cells within 5 h being found for BRD4 **PROTAC 3** by Western blotting; (b) no BRD4 degradation found for caged BRD4 **PROTAC 4** in the absence of light, suggesting installation of NPOM photolabile hindered its binding conformation with E3 ligase; (c) upon irradiation (λ = 365 nm, 6 mins), BRD4 degradation was observed; (d) no change in cellular BRD4 levels observed in the competitive assay (BRD4 **PROTAC-4** compared to with either **MG132** or **MLN4924**), suggesting proteasome- and E3-ligase-mediated degradation of BRD4 protein by BRD4 **PROTAC-4**; and (e) no effect on BRD4 levels observed for nitrosoketone (caging group released fragment)-treated HEK293T cells, suggesting the entire BRD4 protein degradation by light-activated chemical transformed "PROTAC".

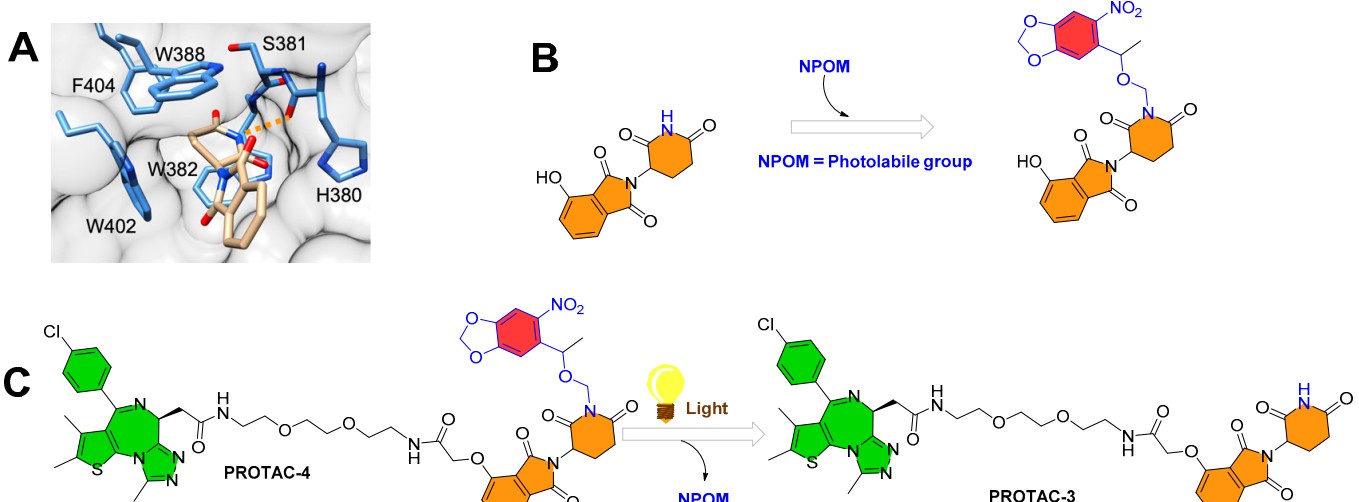

**Figure 5.** (**A**) Chemical design information revealed by X-ray cocrytallise thalidomide structure with CRBN E3-ligase (reprinted (adapted) with permission from [35]; Copyright 2020 American Chemical Society). (**B**) Chemical transformation of thalidomide with NPOM photolabile group. (**C**) Photochemical transformation of **BRD4 PROTAC-4** to activate **BRD4 PROTAC 3**.

To understand the optical control over BRD4 degradation at the cellular level, the authors treated 22Rv1 cells (a castration-resistant prostate cancer cell line) with **BRD4 PROTAC-3** and NPOM-caged **BRD4 PROTAC-4** in the presence/absence of light. Reduc-

tions of 51% and 39% in cell viability were found for BRD4 **PROTAC-3** and NPOM-caged BRD4 **PROTAC-4**, respectively ($\lambda$ = 365 nm, 180 s). In addition, to explore the apoptosis mechanism of BRD4 **PROTAC-4**, caspase-3/7 activation was monitored. BRD4 **PROTAC 3**-treated 22Rv1 cells showed a nearly threefold increase in caspase-3/7 activity, while NPOM-caged BRD4-**PROTAC-4**-treated 22Rv1 cells showed no reasonable effect in the absence of light. However, when irradiated with UV light, caspase-3/7 activity was restored to similar levels to BRD4 **PROTAC 3**.

Collaboratory work by Xue et al. from Peking University and the Southern University of Science and Technology (Shenzhen, China) integrated the photo-control groups into PROTACs to develop photocaged PROTACs (pc-PROTACs) [41]. The Pc-PROTACs remain inactive until irradiated, but upon irradiation with a particular wavelength of light, photochemicals activate the PROTACs with the release of the photolabile group (or photo-control groups). The authors targeted the bromodomain and extra-terminal (BET) protein bromodomain-containing protein 4 (BRD4) on the basis of the chemical design of **dBET1** proposed by the Bradner research group (Dana-Farber Cancer Institute, Harvard Medical School (Boston, USA)), which is molecularly composed of thalidomide (as E3 ligase cereblon ligand) and **JQ1** (a BRD4 ligand) [25]. **dBET1** is a selective BET protein degrader. Numerous in vivo studies showed an efficient photocleavage of 4,5-dimethoxy-2-nitrobenzyl (DMNB) upon irradiation at 365 nm [26]. On the basis of the X-ray co-crystal structure of CRBN-thalidomide (PDB id: 4CI1) [42] and BRD4-JQ1 (PDB id: 3MXF) [43], DMNB was tethered to an amide functionality (-CO*NH*-) of the **JQ1** of **dBET1** to obtain **pc-PROTAC1**, while the imide nitrogen of the thalidomide moiety (CO-*NH*-CO) of **dBET1** was used to form **pc-PROTAC2**, as shown in Figure 6A,B. The photolysis studies by high-performance liquid chromatography (HPLC) showed the generation of **dBET1** upon irradiation (365 nm, 3 mW/cm2) from pc-PROTACs, where quantitative analysis of **pc-PROTAC1** and **pc-PROTAC2** showed a quick photohydrolysis of DMNB with half-times ($T_{1/2}$ = 60 and 105 s, respectively). The study also showed a 50% of **dBET1** from **pc-PROTAC1** (as shown in Figure 6C) and no reasonable amount of **dBET1** produced from **pc-PROTAC2**. Furthermore, the stability of **pc-PROTAC1** was studied in phosphate-buffered saline (PBS) buffer solution for 24 h in dark conditions, where $\approx$88% of **pc-PROTAC1** and **no dBET1** release were observed. These studies led the authors to examine the **pc-PROTAC1** for the photocaging PROTAC mechanism [41].

As anticipated, an increase in the molecular size of **pc-PROTAC1** reduced approximately 100 and 300 times its BRD4-binding affinity ($IC_{50}$ = 7.6 $\mu$M) when compared to the **JQ1** ($IC_{50}$ = 71 nM) and **dBET1** ($IC_{50}$ = 22 nM), respectively. Ramos cells were used to study the dose-dependent BRD4 protein degradation of **pc-PROTAC1**. The **pc-PROTAC1** at 3 $\mu$M did not show a reasonable degradation, but upon UV irradiation ($\lambda$ = 365 nm for 3 min), a dose-dependent BRD4 protein degradation was observed. A comparative degradation profile of 0.3 $\mu$M **pc-PROTAC1** was found to be equivalent to 0.1 $\mu$M **dBET1**. A maximum Brd4 degradation efficacy (Dmax) of 93% was achieved with 1 $\mu$M **pc-PROTAC1**. Later, irradiation time intervals were evaluated, where exposure of 0.3 min (at 365 nm) reduced the BRD4 protein cellular levels, while prolonged exposure (time = 3 min) completely degraded the BRD4 protein cellular level [41].

To expand the antiproliferative spectrum of **pc-PROTAC1**, various concentrations of **pc-PROTAC1** and **dBET1**, with or without irradiation on BRD4-dependent human Burkitt's lymphoma cells (Namalwa cells), were examined. Interestingly, without irradiation, **pc-PROTAC1** ($GI_{50}$ = 3.1 $\mu$M) showed a weaker antiproliferative activity than **dBET1** ($GI_{50}$ = 0.34 $\mu$M) and was unable to achieve reasonable antiproliferation, even at 50 $\mu$M (**pc-PROTAC1**). However, upon irradiation, **pc-PROTAC1** showed a comparative antiproliferative activity ($GI_{50}$ = 0.4 $\mu$M) compared to **dBET1**. Furthermore, a 10-day colony-forming assay was investigated on a liver cell line (HUH7 hepatocellular carcinoma cells) to assess the long-term antiproliferative effects of **pc-PROTAC1**. Without irradiation, **pc-PROTAC1** exhibited no change in colony density, while upon irradiation, **pc-PROTAC1** (at 5 $\mu$M) achieved a maximum inhibition. These studies showed that UV-light-based

intracellular photoactivation of **pc-PROTAC1** into a PROTAC form (**dBET1**) degrades the BRD4 cellular protein level in cancer cells [41].

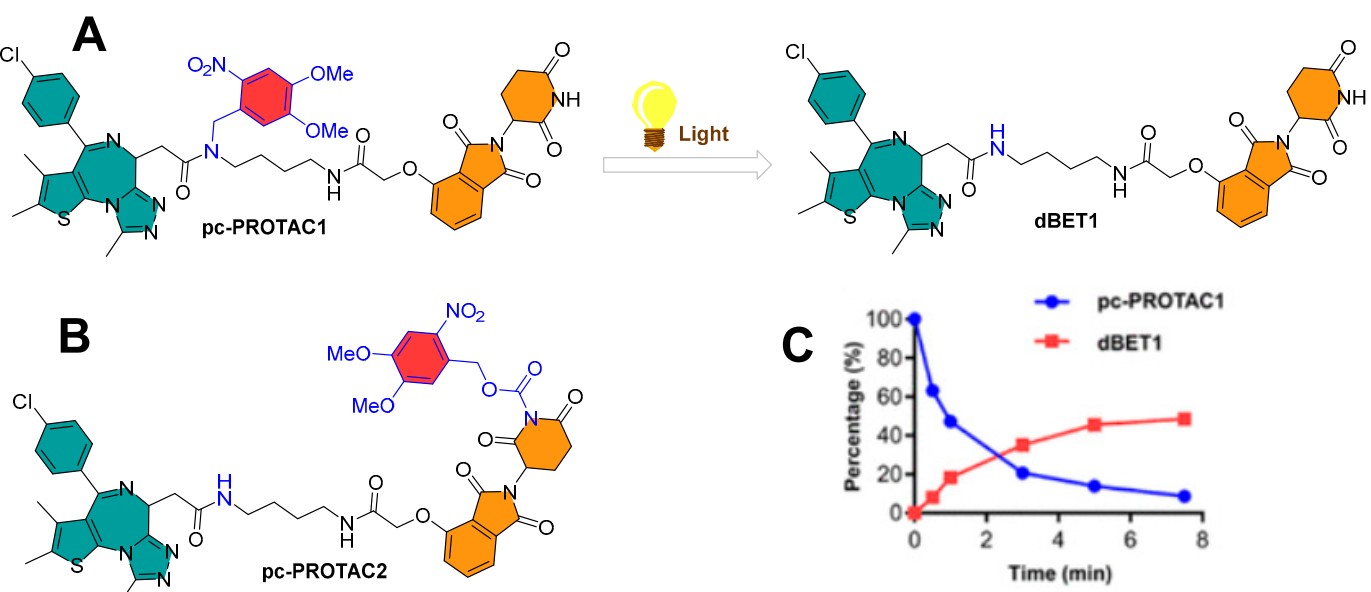

**Figure 6.** (**A**) Light-activating BRD4 protein photocaged PROTAC (**pc-PROTAC1**) into **dBET1**. (**B**) Chemial structure of pc-PROTAC2. (**C**) Light-induced release efficiency of **pc-PROTAC1** to generate dBET1 (reprinted (adapted) with permission from [41]; Copyright 2019 American Chemical Society).

The authors explored in vivo activity of **pc-PROTAC1** using zebrafish as a model organism. Structurally, the BRD4 protein has interspecies conserved domains (domains: BD1, BD2, and ET) [44], where the primary binding site of JQ1 is characterized by BD1 and BD2 domains, which are importantly highly conserved between zebrafish and humans [43]. Additionally, CRBN of zebrafish has a high sequence similarity with the human ortholog protein. Zebrafish embryo has a higher level of BRD4 [44], which led the authors to use 12 h post-fertilization zebrafish embryos. Those embryos were treated with **pc-PROTAC1** (100 μM in embryo medium), irradiated (λ = 365 nm, for 10 min), and incubated at 28.5 °C. The following samples were prepared: DMSO (1% in embryo medium)-treated embryos as a blank control, **dBET1** (50 μM in embryo medium)-treated embryos as a positive control, and **pc-PROTAC1** (100 μM in embryo medium)-treated embryos with no irradiation as a negative control. **pc-PROTAC1** with irradiation or **dBET1**-treated embryos showed a thinner yolk extension with respect to both the controls (DMSO and **pc-PROTAC1** without irradiation) at 24 h post-fertilization. Quantitative cellular BRD4 protein levels were studied by Western blotting from whole-cell extracts of the treated embryos at 36 h post-fertilization to evaluate the degradation activity of **pc-PROTAC1**. In the study, substantial BRD4 protein degradations in embryos treated with **dBET1** and **pc-PROTAC1** (50 or 100 μM) with irradiation were observed, demonstrating the light-activating degradation activity of **pc-PROTAC1** in zebrafish [41].

*3.3. Photocaging PROTACs of BTK Kinase Proteins*

The same authors (Xue et al. from Peking University and the Southern University of Science and Technology (Shenzhen, China)) explored photocaged-PROTAC to degrade the BTK protein. They used **MT-802** as a PROTAC template, and DMNB was tethered to the imide nitrogen of **MT-802** to form **pc-PROTAC3** (as shown in Figure 7A). **MT-802** is a PROTAC structure composed of **ibrutinib** as a BTK protein inhibitor and CRBN E3 ligase ligand, which was developed by Crews' group (Yale University, New Haven, Connecticut, USA) to degrade the BTK protein [45]. A similar profile (absorption spectrum, stability,

and photo-induced release efficiency) of **pc-PROTAC3** was found in comparison to **pc-PROTAC1**. The IC$_{50}$'s of (**pc-PROTAC3**, **MT-802**, **ibrutinib**) against BTK were investigated by HTRF KinEase assays. Various concentrations of **pc-PROTAC3** were used to treat Ramos cells in the absence/presence of light (as shown in Figure 7B). However, a significant dose-dependent reduction in BTK levels was observed upon irradiation (at 365 nm for 3.5 min), as shown in Figure 7C [41].

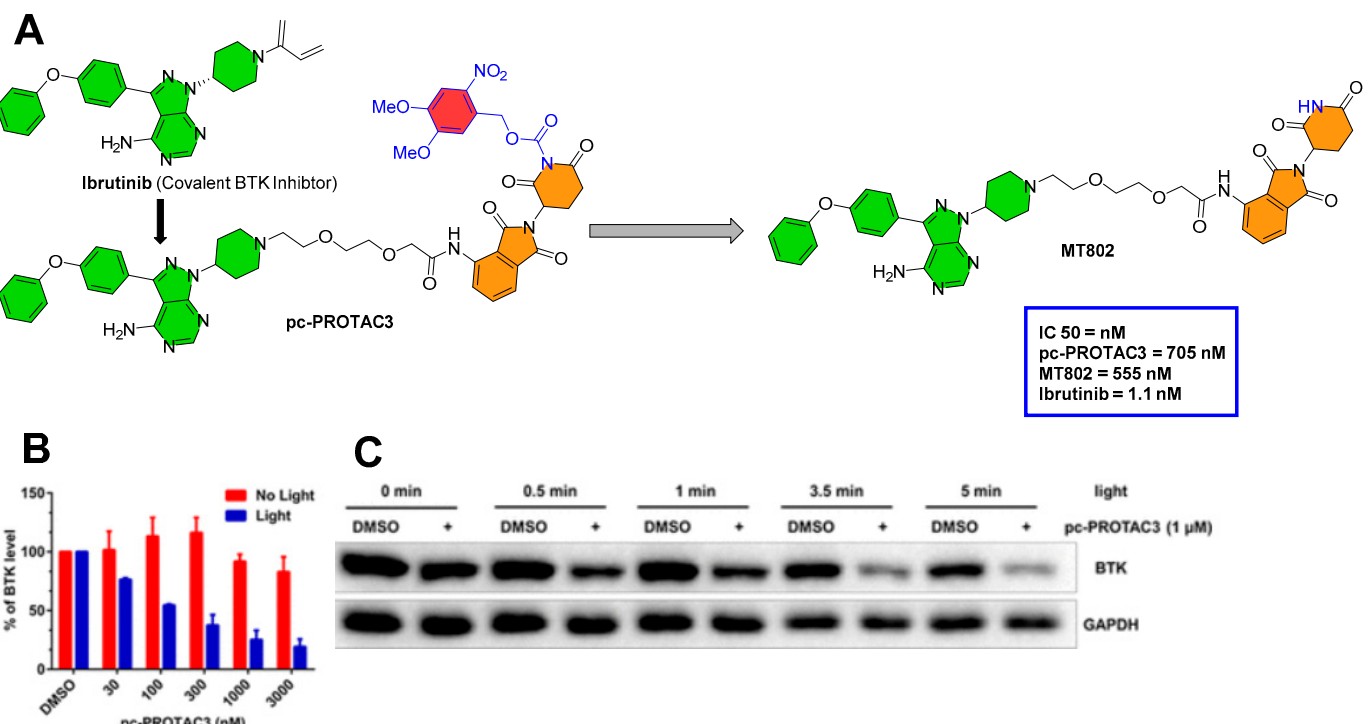

**Figure 7.** (**A**) Chemical design transformation of pc-PROTAC3 from ibrutinib. (**B**) Upon irradiation, dose-dependent reduction of cellular BTK levels by pc-PROTAC3. (**C**) Western blot assay result of the BTK levels at various irradiation times in Ramos cells. Subfigures (**B**,**C**) reprinted (adapted) with permission from [41]; Copyright 2019 American Chemical Society.

## 4. Photoswitches in PROTACs

Despite the implementation of the photochemical elements of photopharmacology into PROTACs to design photocaged PROTACs, there is an intrinsic flaw with their *irreversible activation of the pharmacophore from PPG*; therefore, their systemic clearance entirely depends on the cellular metabolism. Thus, the integration of photoswitches (as illustrated in Figure 8) was invented as another strategy of photopharmacology that amplifies the photochemical control of PROTACs over targeted protein degradation (as shown in Figure 9). Furthermore, the reversible nature of photoswitches provides a spatiotemporal handle to the researchers for controlling protein degradation with more precision. In photoswitchable PROTACs, a specific wavelength of light prefers one isomer specificity (either *cis* or *trans*), which could be an active or inactive form. In contrast, quantum yield and photoisomeric equilibrium can be easily shifted in either direction that activates or stops protein degradation [46].

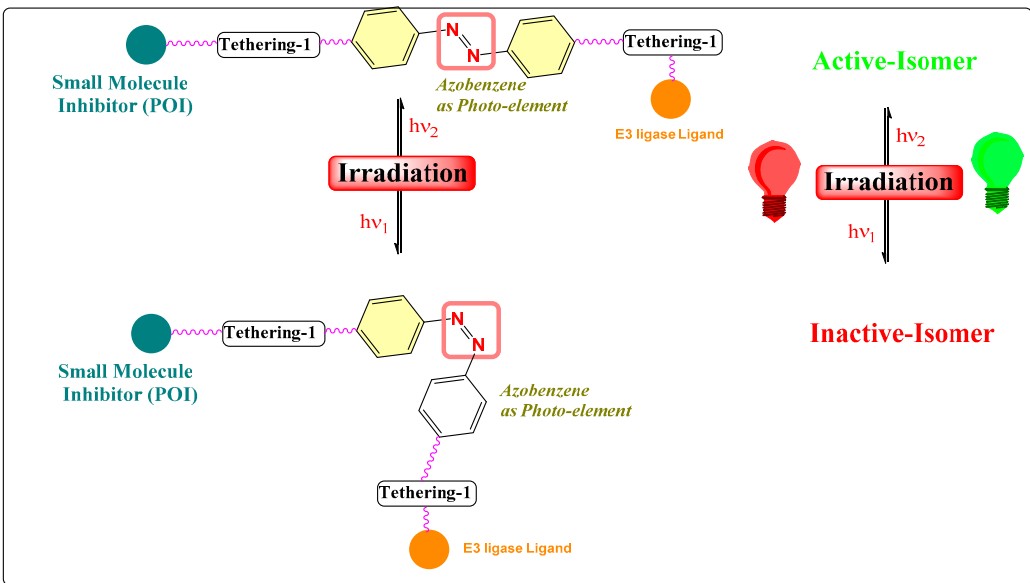

**Figure 8.** Photoswitchable PROTACs: Light-activating azoswitch brings the conformation change in the PROTAC structure, either facilitating its availability for simultaneous binding to POI and E3 ligase or shortening the linker length required to reach both proteins (POI and E3 ligase) at the same time. Such conformational changes, typically display a reversible configurational isomerization with the corresponding activity–inactivity towards specific protein degradation [47].

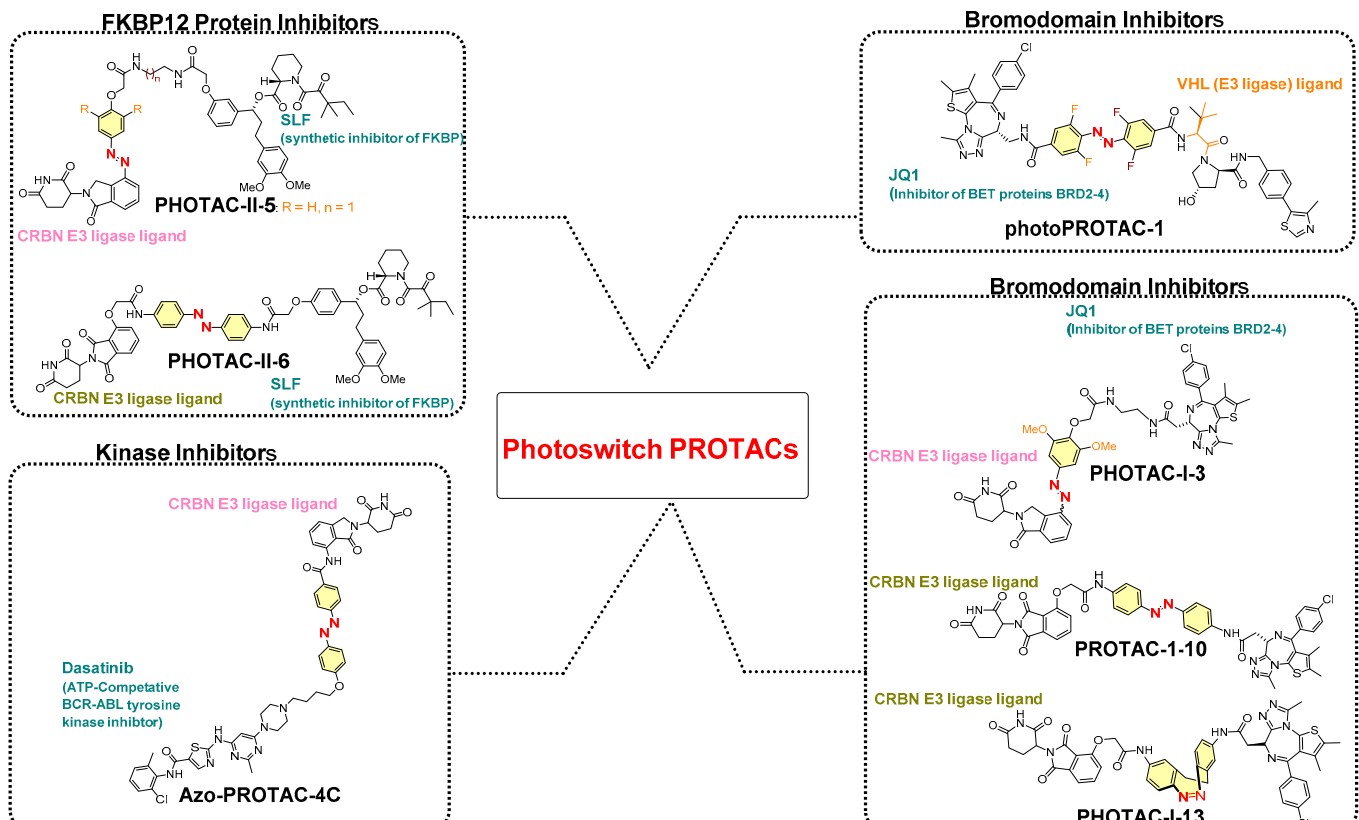

**Figure 9.** Schematic representation of reported Photoswitchable PROTACs. The figure was reproduced with the permission of Negi et al. [47]; Copyright © 2022 Chemistry Europe.

Previous studies on azobenzene-based reversible photoswitchablity and their photochemical characterization encouraged researchers to incorporate them into PROTAC struc-

tures. Most of the reported photoswitchable PROTACs are azobenzene based, except for **PHOTAC-I-13** (the only photoswitchable PROTAC where diazocines act as photoswitches in place of azobenzene, as shown in Figure 12). The relatively high use of azobenzene-based photoswitch directs its essential photochemical qualities: (a) fatigue resistance, (b) facile control over (*E*)/(*Z*)-geometrical conversion, (c) easy functionalization of the aromatic rings improving/modifying photothermal properties on the basis of requirements of a chemical biologist or medicinal chemist, and (d) relatively smaller size that does not add a significant proportion to the molecular obesity of the overall molecule [48–50]. The incorporation of azobenzene leads to two configurational isomers of PROTAC, where the bioactive photoisomeric isomer could be either a thermodynamically more stable (*trans or E*) isomer (such as **azo-PROTAC4C**, **photoPROTAC-1**) or a metastable (*Cis-Z*) isomer (PHOTACs). On the basis of studies claiming photoswitchable PROTAC, the following structure–activity relationship (SAR) can be drawn:

*(A) Rigidifying the azobenzene photoswitch* into one configuration form. Rigidification of an azobenzene-based photoswitch can be achieved either by macrocyclization [51] or by replacing azobenzene with similar ring structures (such as diazocine ring [50]). However, no photochemical control over POI degradation was found in the case of diazocine-containing PROTAC (**PHOTAC-I-13**), showing the unsuitability of **the** diazocine-based photoswitch compared to a similar PROTAC structure with an azobenzene photoswitch (such as **PHOTAC-I-10**, which showed degradation of the targeted protein, as shown in Figure 12).

*(B) Positioning of photoswitch in PROTAC structure*: (i) if it is positioned adjacent to the E3 ligase warhead, it will not influence PROTAC's ability to adopt the binding conformation for the targeted protein. Still, its proximity to the E3 ligase warhead alters the binding conformation of PROTAC to its E3 ligase. Therefore, varying degrees of targeted protein degradation activity were reported (such as similar targeted protein degradations being achieved for **PHOTAC**-I-1 to **8**, while no protein degradations for **PHOTAC-I-9** were measured, as illustrated in Figure 12).

(i)   Photoswitch positioning in the linker region, equidistant from both sides of the PROTAC structure. This is a typical strategy implemented to find an optimum length of the linker, besides improving the chemical nature and orientation of the PROTAC. Reported photoswitchable PROTACs with such positioning of azobenzene photoswitch are commonly tethered through linear hydrocarbon chains (polyether, aliphatic, or amide-aliphatic linkage) to both terminals (for example, **Azo-PROTAC-2C-6C** shown in Figure 11, **PHOTAC-I-11** and **12** shown in Figure 12, *Cis-/trans-***PhotoPROTAC-1** shown in Figure 13, and **PHOTAC-II-6** shown in Figure 14).

(ii)  If the photoswitch is directly tethered to the POI warhead, it could restrict the binding conformational of respective photoswitchable PROTAC to its POI. However, it would also pose a synthetic challenge to an organic chemist in building such molecular structures where the azoswitch tethers POI warheads and could significantly reduce the targeting protein degradation; therefore, such attempts were not reported in the literature.

### 4.1. Photoswitchable PROTACs Targeting Kinase Protein

Jin et al., from the School of Pharmacy at China Pharmaceutical University, Nanjing (China) [48], studied the binding conformation of lenalidomide from an X-ray co-crystal structure (PDB id: 4TZ4 [52]; as can be seen in Figure 10A) to design Azo-PROTACs.

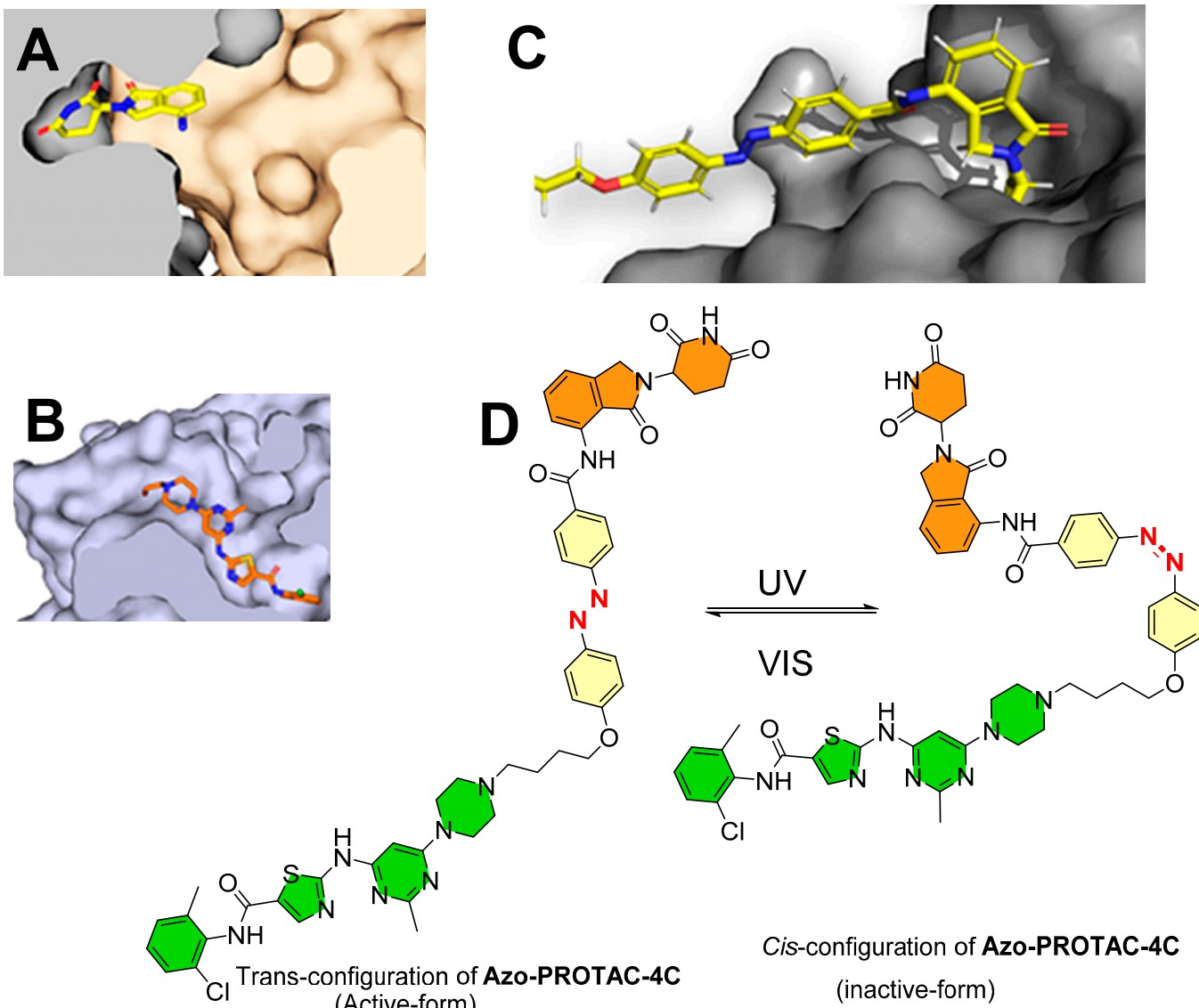

**Figure 10.** (**A**) X-ray structure of *Homo sapiens* cereblon with DNA-damage-binding protein 1 and lenalidomide (PDB id: 4TZ4) [52]. (**B**) X-ray structure of ABL protein with **dasatinib** (PDB id: 2GQG) [53]. (**C**) Docking of **Azo-PROTAC-4C-*trans*** and CRBN. Subfigures (**A**–**C**) were adapted with permission from Jin et al. [48]; Copyright © 2020 American Chemical Society.

On the basis of the ligand–protein interaction of lenalidomide with E3 ligase, the binding pocket was relatively smaller and possessed a steric effect at the solvent boundary interface. On the basis of these observations, the authors rationalized incorporating an azobenzene photoswitch at the C3-position of the phenyl in lenalidomide. To develop a photoswitchable PROTAC against BCR-ABL fusion and ABL protein, **dasatinib** was selected as a POI warhead. **Dasatinib** is clinically used in acute lymphoblastic leukemia (ALL), chronic myelogenous leukemia (CML), and in CML patients with acquired **imatinib** resistance. It is a second-generation ATP-competitive tyrosine kinase inhibitor of BCR/ABL proteins. On the basis of the conformational binding of **dasatinib** with ABL protein in an X-ray co-crystal structure (PDB id: 2GQG [53]; as shown in Figure 10B) and molecular docking of azobenzene-incorporated PROTACs with BCR-ABL fusion protein, these biophysical studies illustrated that only *trans*-form was combinatory due to steric (Figure 10C [48]), suggesting that the *cis*- and *trans*-forms could bring significant differences in the degradation activity of targeted proteins. To find a suitable linker length, the authors performed a chemical screening, wherein the resulting PROTACs were tested for their degradation (ABL

and BCR-ABL) activity on the K562 myelogenous leukemia cell line (ABL, BCR-ABL, and E3 ligase CRBN expressed cell line). Among all the synthesized photoswitchable PROTACs (**Azo-PROTACs-2C** to **6C**), **Azo-PROTAC-4C** had an appropriate linker length with higher BCR-ABL fusion protein degrading potency, as shown in Figure 11A.

## A. Chemical Design

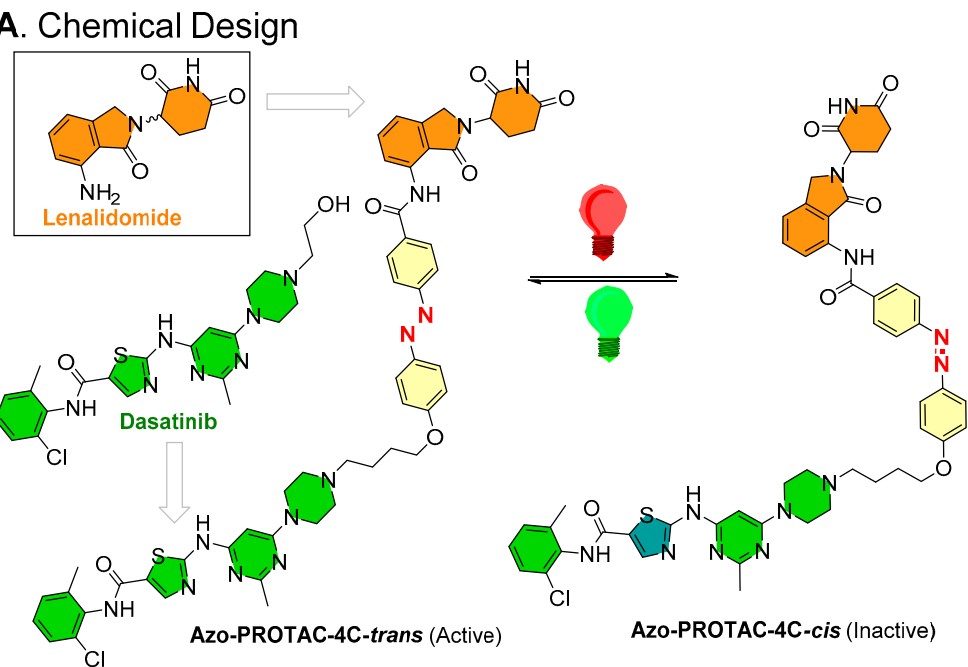

## B. Screening of Linker Length

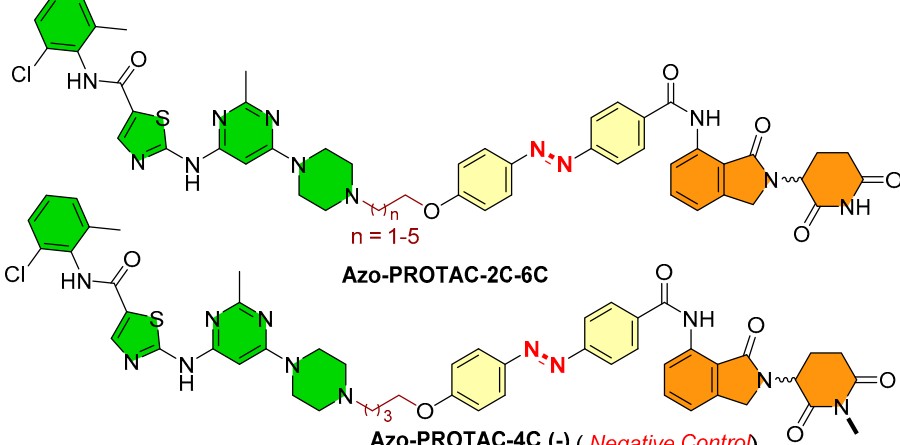

**Figure 11.** (**A**) Chemical design for Azo-PROTACs. (**B**) Linker length screening for Azo-PROTACs.

By UV–VIS spectroscopy photoisomerization, kinetic studies were performed to measure the photochemical control and stability over reverse switching of **Azo-PROTAC-4C**. The *trans*-configuration of **Azo-PROTAC-4C** showed $\lambda_{max}$ at 361 nm. When **Azo-PROTAC-4C** was exposed to UV-C light, varying degrees of reduction of peak ($\lambda_{max}$ = 361 nm) were recorded, indicating **Azo-PROTAC-4C** conversion into *cis*-configuration. Later, 1 h exposure was found to be sufficient to convert *trans*-configuration into *cis*-configuration. Further exposure with white light to **Azo-PROTAC-4C** showed *cis*-configuration within 4 h. **Azo-PROTAC-4C** is a T-type photoswitch [54], and a half-life of spontaneous thermal relaxation ($T_{1/2}$ = 620 min at 25 °C) was measured in dark conditions. In cellular assays, **Azo-PROTAC-4C** showed cell-selectivity against BCR-ABL-driven K562 cells (EC$_{50}$ = 28 nM in

cell viability assay, $IC_{50}$ = 68 nM in cellular proliferation assay), while no activity was found for non-BCR-ABL-derived cell lines (HCT116 colorectal carcinoma cells, A549 pulmonary carcinoma cells, and MCF-7 cells). Furthermore, negative control of **Azo-PROTAC-4C** was developed to determine the mechanism of targeted protein degradation. Methylation of imide *(-NH-)* of glutarimide (in the lenalidomide) of **Azo-PROTAC-4C** led to the formation of **PROTAC-4C (-)** (as a negative control) (as illustrated in Figure 11B). **Azo-PROTAC-4C** (-) showed no BCR-ABL and ABL protein degradations, suggesting that BCR-ABL and ABL degradation of **Azo-PROTAC-4C** is CRBN-mediated ubiquitination. Further use of **dasatinib** (as a POI inhibitor), lenalidomide (as an E3 ligase ligand), and an inhibitor NEDD8-activating enzyme (**MLN4924**) in competitive assays, and observing no degradation of BCR-ABL, suggests the **Azo-PROTAC-4C** degradation activity (a) utilizes the active site of BCR-ABL (in case of dasatinib), (b) is cereblon-E3 ligase mediated (in the case of lenalidomide), and (c) depends upon NEDD8-activating enzyme (in case of **MNL4924**). All these outcomes show confirmation that **Azo-PROTAC-4C** is a CRBN-dependent ABL degrader.

Interestingly, a substantial difference in ABL and BCR-ABL degradation were recorded for *cis-* and *trans-*configurations of **Azo-PROTAC-4C**. In a dose–effect evaluation, a slight degradation at 25 nM but a significant BCR-ABL and ABL protein degradation at 100 nM was observed for *trans-*configuration of **Azo-PROTAC-4C**. However, no noticeable BCR-ABL degradation was observed, even at 500 nM for cis-configuration of **Azo-PROTAC-4C**, which could have been due to the "*hook effect*" (a typical limitation with PROTAC approaches) or based on the presence of photostationary state of *cis-*configuration of **Azo-PROTAC 4C** in the solution [46]. The spatiotemporal degradation of ABL and BCR-ABL proteins by **Azo-PROTAC-4C** was studied by using time-course analysis. In this study, *trans-*configuration of **Azo-PROTAC-4C** showed ABL protein degradation started after 4 h treatment, significant degradation of BCR-ABL and ABL proteins after 10 h, and > 90% of BCR-ABL degradation after 32 h treatment. Interestingly, no noticeable BCR-ABL reduction for *cis-*configuration of **Azo-PROTAC-4C** was recorded, even after 32 h of treatment, which was somewhat surprising because of the reported half-life ($T_{1/2}$ = 620 min at 25 °C) and, therefore, a large proportion of *trans-*configuration of **Azo-PROTAC-4C** could be present at earlier time points [46].

To evaluate the reversible photoswitchablity of **Azo-PROTAC-4C** against its protein degradation activity, K562 cells were treated with *trans-*configuration of **Azo-PROTAC-4C** for 24 h and then irradiated with UV-C light every 4 h. ABL and BCR-ABL levels were increased over time in the UV-irradiated group, while low cellular levels of these proteins were maintained in the visible light group. Later, after preincubation of K562 cells with *cis-*configuration of **Azo-PROTAC-4C** shielded from light, K562 cells were transferred to a fresh medium and irradiated with visible light, where BCR-ABL degradation was observed with time. However, evaluation of the cell viability of *cis-configuration of* **Azo-PROTAC-4C** would be quite interesting in order to engage researchers, although the authors did not perform this [46].

*4.2. Photoswitchable PROTACs in Bromodomain Proteins*

A collaboration of the Trauner research group (Department of Chemistry, New York University, NY, USA) and the Pagano research group (Department of Biochemistry and molecular pharmacology, New York School of Medicine) incorporated an azobenzene photoswitch to form photoswitchable PROTACs, called PHOTACs. In these photoswitchable PROTACs, **(+)-JQ1** (a high-affinity inhibitor of BET proteins BRD2-4) was selected as a POI inhibitor, alongside two different types of E3-ligase ligands (with lenalidomide: **PHOTAC-I-1** to **8**, and pomalidomide **PHOTAC-I-9** to **13**) [50], to target CRBN E3 ligase, as shown in Figure 12. Among azobenzene-photoswitch PROTACs (**PHOTAC-I-1-12**), diazocine-based photoswitch PROTACs (**PHOTAC-I-13**) were also designed to degrade the bromodomain proteins, where **PHOTAC-I-3** emerged as a lead compound. The **PHOTAC-I-3** switched to the *cis-*configuration at a certain wavelength (λ = 390 nm, PSS > 90%) and achieved

similar photostationary states (PSSs) at a range of 380–400 nm. An efficient *trans*-to-*cis* configuration of **PHOTAC-I-3** photoisomerization was observed at 370 and 450 nm, while in the absence of light *cis*-configuration of **PHOTAC-I-3, it** gradually relaxed back to *trans*-configuration of **PHOTAC-I-3** ($T_{1/2}$ = 8.8 h at 37 °C in DMSO). However, when irradiated ($\lambda \geq$ 500 nm), the chemical equilibrium shifted to *trans*-configuration of **PHOTAC-I-3** ($\lambda \geq$ 450 nm, PSS of >70%).

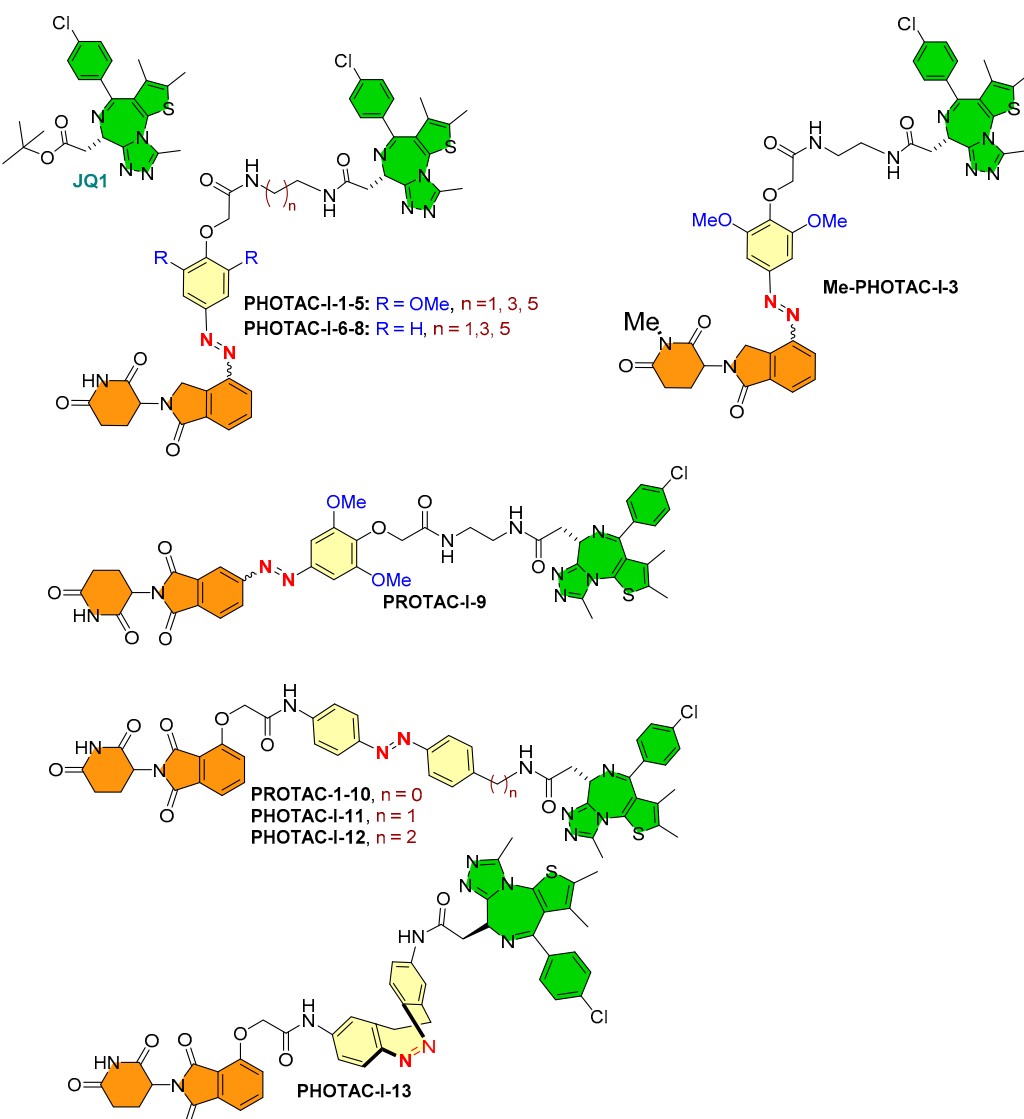

**Figure 12.** PHOTACs: PROTACs based on azobenzene photoswitches (**PHOTAC-I-1-12**) and PROTACs based on diazocine photoswitch (**PHOTAC-I-13**) for bromodomain proteins (BRD2, BRD3, and BRD4).

To evaluate the optical qualitative nature of these photoswitchable PROTACs (**PHOTAC-I-1** to **13**), human acute leukemia cell lines (RS4;11) in a 96-well plate were treated with increasing concentrations of these PROTACs. Later, they were irradiated with either 390 nm light pulses (100 ms/10 s) for 72 h or as a control incubated in dark conditions. Interestingly, **PHOTAC-I-3** showed an optical response where a sevenfold difference was measured in median effective concentration ($EC_{50}$): $EC_{50}$ = 88.5 nM when irradiated with 390 nm light, whereas $EC_{50}$ = 631 nM in the dark conditions. Moreover, similar optical responses were noted for **PHOTAC-I-1**,**2**,**4** to **8**,**10**. Interestingly, **PHOTAC-I-9, 11** to **13,** and **(+)-JQ1** (**(+)-JQ1** was used as a control in these experiments) did not show differences in their $EC_{50}$ from irradiated samples against those samples incubated in the dark, suggesting a role

of E3-ligand chemical specificity as **PHOTAC-I-1** to **8** (having lenalidomide chemicals), as opposed to **PHOTAC-I-9** to **13** (having pomalidomide as E3 ligand chemical structural features).

To understand light-dependent targeted BET protein BRD2-4 degradation for these PROTACs, Western blotting was used to determine the cellular protein levels. The authors treated RS4;11 cells with increasing concentrations of **PHOTAC-I-3** for 4 h and pulse irradiation (λ = 390 nm, pulse = 100 ms/10 s). A significant decrease in BRD4 levels when irradiated (λ = 390 nm), but not in the dark, was observed for **PHOTAC-I-3** in the range of 100 nm to 3 μM. At 10 μM, a "*hook effect*" was observed, a common issue with PROTAC strategies [17,55,56]. Similar cellular protein level reduction was also noted for BRD3 at 100 nm to 3 μM under violet light irradiation but not in the dark. Conversely, a moderate BRD2 degradation was observed in a narrower concentration range. To characterize the degradation mechanism of these photoswitchable PROTACs, several biochemical experiments were conducted using competitive assays. Upon irradiation, **PHOTAC-I-3** (1 μM) was used along with CRL inhibitor (**MLN4924** at 2.5 μM). **MLN4924** inhibits neddylation and, subsequently, cellular CRL (including CRL4$^{CRBN}$) activities. Therefore, BRD2–4 cellular level recovery was observed in this assay, suggesting a CRBN-dependent degradation. Similar results were also obtained from small interfering RNA (siRNA) knockdown of CRBN and methylation of the glutarimide [57].

To expand the optical responsive into different cancer cell types, light-dependent targeting BRD4 degradation of **PHOTAC-I-3** was evaluated in two breast cancer cell lines (MB-MDA-231 and MB-MDA-468).

To access the reversible photoswitchablity of **PHOTAC-I-3**, a rescue experiment was performed. Continuous irradiation for an initial 1 min to **PHOTAC-I-3** with λ = 390 nm (activating wavelength), followed by pulse irradiation with λ = 525 nm (deactivating wavelength), resulted in an initial decrease in cellular BRD2 levels, but they recovered rapidly, even compared to when it was left in the dark conditions.

One key characteristic of photopharmacology is *color dosing* [58,59]. Color dosing is a photocontrol ability where the color intensity of the incident light can be compared with the respective concentration of the active isomer [60]. The color strength of the incident light can be used to estimate the extent of targeted protein degradation, where it is informative of the ratio of active *and* inactive isomers in the photostationary states [61]. In cell viability assays, left-shifted curves were observed, where the color steadily approached 390 nm. Conversely, Western blotting showed maximum degradation of BRD4 at λ = 390 nm, while BRD4 levels recovered upon increasing the incident light wavelength, suggesting a reversible photoswitchablity over BRD4 protein degradation of **PHOTAC-I-3**.

Crews (from Yale University, USA) and Carreira (Laboratory of Organic Chemistry, ETH Zürich, Switzerland) and their coworkers rationalized the linker region of PRO-TACs [49]. Previous biophysical studies highlighted the small changes in length and chemical nature of the linker region affect the ternary complex formation [62,63]. A sufficient length of linker is required for PROTACs so that both of the warheads are able to rotate and fit into the recruited protein (POI and E3 ligase) to form a ternary complex [64]. In the reported examples, the critical difference linker's length between inactive and active was calculated 3 Å, which interestingly matched with typical distances (3–4 Å) reported with *cis-* and *trans*-azobenzene photoswitches [45,65]. On the basis of these key structural features, the authors were prompted with an idea "of design of a *trans*-configuration of photoPRO-TAC that will ideally have an optimized linker length for efficient ternary complex while *cis*-configuration of photoPROTAC would have a shorter distance to reach the binding pocket of the second protein partner (which could be E3 ligase or POI)". Therefore, the authors chemically transformed the polyether linker region of **ARV-771 (ARV-771** is a PRO-TAC of bromodomain proteins, as shown in the structure in Figure 13A) into azobenzene photoswitch in order to design a photoswitchable PROTAC (photo-PROATCs) [49].

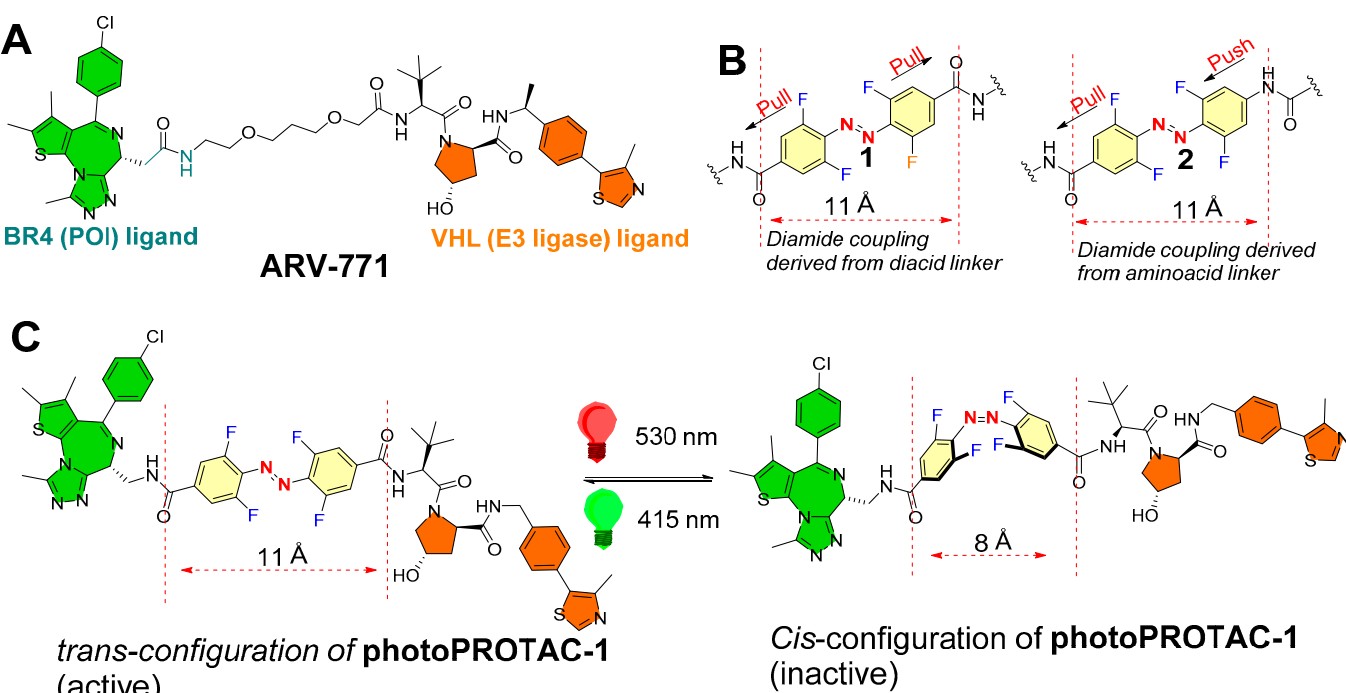

**Figure 13.** (**A**) Structure of **ARV-771**, a BET PROTAC. (**B**) Molecular azoswitch frames: *o*-F₄ azobenzene modular switches as "pull–pull" and "push–pull". (**C**) Derivatized photoswitchable PROTAC with "pull–pull" diacid azoswitch.

Secondly, the authors envisioned an ideal azobenzene photoswitch having prolonged photostationary states (≈days) that are populated during initial irradiation stimuli and must persist throughout the experiment so that continued pulse irradiation exposure can be avoided. This required an azobenzene whose *cis*-configuration shows configuration isomerization stability in biological settings, which led the authors to introduce bistable ortho-tetrafluoroazobenzenes (*o*-F₄-azobenzenes) as azobenzene photoswitches [66], which had reported thermal $\tau_{1/2}$'s ≈ 2 years at 25 °C as opposed to a few hours, as shown by its parent *cis*-azobenzenes [67]. The authors replaced the polyether linker from **ARV-771** skeleton with diamide azobenzene photoswitches, *o*-F₄-azobenzene diacid (1, representing the *pull–pull* azoswitch mechanism, Figure 13B), and *o*-F₄-azobenzene amino acid (2, representing the *push–pull* azoswitch mechanism, Figure 13B).

The authors studied the photochemical properties. The photochemical properties of *pull–pull* azoswitch-based **photoPROTAC-1** was studied in DMSO. Irridation (λ = 415 nm) was found to be efficient in producing the photostationary state of 95% for *trans*-configuration of photoPROTAC-1 (separation was performed with the help of HPLC). However, irradiation (λ = 530 nm) showed the photostationary state of 68% for *cis*-configuration of **photoPROTAC-1**, which is comparatively lower than examples of *o*-F₄-azobenzenes with higher *cis*-isomer photostationary states [67,68]. From the spectra, pure *cis*-configuration, and *trans*-configuration of **photoPROTAC-1**, n–π* absorption maxima of both isomers were found with a difference of 47 nm. The quantum yields for both isomerization reactions (ϕEZ (530 nm) = 0.28, ϕZE (415 nm) = 0.65) were found in a similar order to previous literature for underivatized *o*-F₄-azobenzenes [67], suggesting an efficient isomerization upon photon absorption. Moreover, *pull–pull* ortho-F4 azobenzene reported bistability was retained in the photoPROTAC-1, and no reverse thermal isomerization of *cis*-configuration was noticed (DMSO, acetonitrile, aqueous buffer at 37 °C, even for several days), ensuring the suitability of a *pull–pull* azoswitch as photoswitches for PROTAC strategies.

Furthermore, a stability retention was noticed with a photoPROTAC-1 (50 μM) in reduced glutathione (10 mM) over 72 h, demonstrating its suitability for biological studies. To ensure stability in cellular systems, cells were incubated without or with photoPROTAC-1

(25 μM). Later, cells were harvested and lysed in phosphate-buffered saline by multiple freeze–thaw cycles in different time periods up to 24 h, and lysates were studied by LC–MS for reduced forms as well as non-reduced forms of **photoPROTAC-1**. During data processing, fragment masses matched with reduced form remained undetected, while isotopic mass distribution after 24 h matched with a nonreduced form of the **photoPROTAC**-1, assuring physiological stability of the **photoPROTAC-1**.

For biological testing, Ramos cells were used. For initial 20 min irradiation with (λ = 415) and (λ = 530 nm), **PhotoPROTAC-1** solutions were used to obtain trans-configuration and cis-configuration, respectively. Then, 1 min of holding took place, followed by brief shaking (vortexing) and irradiating again for the next 10 min. *Trans*-configuration and *cis*-configuration of **photoPROTAC-1** solutions were diluted to the obtained the required molar concentrations, which were then added to Ramos cells. Incubating Ramos cells with a nanomolar concentration of trans-configuration of **photoPROTAC-1** for 6.5 h induced a significant BRD2 degradation, while cis-photoPROTAC-1 failed to induce BRD2 degradation in the tested range of molar concentrations. However, a prolonged incubation of 18 h with *cis*-configuration or *trans*-configuration of **photoPROTAC-1** failed to alter the BRD2 degradation efficiencies. Importantly, **photoPROTAC-1** did not degrade the BRD2 cellular level and required an initial 415 nm irradiation, suggesting a mixture of trans-configuration and cis-configuration of photoPROTAC-1 present in the solution. The most plausible explanation for such an observation would be the higher binding affinity of cis-configuration for either BRD2 or VHL compared to trans-configuration of **photoPROTAC-1**, which prevents binding of the active trans-isomer and therefore abolishes the degradation.

To study the time-dependent degradation profiles of trans-configuration and cis-configuration of **photoPROTAC-1**, various incubation time points (1–24 h) were selected with Ramos cells. No reasonable BRD2 degradation of trans-configuration of **photoPROTAC-1** was observed within 1 h, nor after 3 h. Conversely, degradation increased in the next 4 h, reached a maximum after 7 h, and remained unchanged for 17 h. A concomitant targeting of trans-configuration of **photoPROTAC-1** with **MLN-4924** (selective NEDD8 inhibitor) showed a loss of BRD2 degradation potency, suggesting the proteasomal-mediated protein degradation activity.

Interestingly, *cis*- or *trans*-configuration of photoPROTAC-1 did not show a substantial BRD4 degradation, while **ARV-771** was found to be an efficient degrader of BRD4 as well as BRD2. This observation might be explained on the basis of the structural differences of **ARV-771** and **photoPROTAC-1**, which affected their ternary complex stability for BRD4 protein degradation: (a) the presence of a reversed amide bond between **JQ-1** and o-$F_4$-azobenzene moiety in **photoPROTAC-1** compared to polyether linkage of **ARV-771**, and (b) a higher intrinsic rigid character of azobenzene during their incorporation into **photoPROTAC-1** [69].

To evaluate the reversible photoswitching between *cis*-configuration and *trans*-configuration of **photoPROTAC-1**, two test tubes were irradiated by 415 and 530 nm LEDs for 30 mins to obtain *trans*-configuration and *cis*-configuration of **photoPROTAC-1,** respectively. *Trans*-configuration of **photoPROTAC-1** and *cis*-configuration of **photoPROTAC-1** were then subdivided (1:1) into two tubes each, whereas parent test tubes were stored separately. The new two test tubes were irradiated: the test tube that contained trans-configuration of photoPROTAC-1 was irradiated with λ = 530 nm, while the test tube contained *cis*-configuration of **photoPROTAC-1** was irradiated with λ = 415 nm. Varying concentrations of singly irradiated (415 nm, trans-photoPROTAC-1; or 530 nm, cis-photoPROTAC-1) or doubly irradiated (530/415 nm, trans-photoPROTAC-1; or 415/530 nm, cis-photoPROTAC-1) **photoPROTAC-1** solutions were incubated with Ramos cells to study the reversible photoisomerization. The treated cells were lysed, and the cellular levels of BRD4 and BRD2 were estimated after 18 h. Interestingly, a similar pattern of BRD2/4 degradation was noticed for singly irradiated *cis*-configuration and *trans*-configuration of **photoPROTAC-1,** and reversed protein degradation activity was observed in doubly irradiated treatments, ensuring the reverse photoswitchablity of **photoPROTAC**.

The spatiotemporal control of **photoPROTAC-1** was evaluated. Two sets of *trans*-configuration of photoPROTAC-1-treated Ramos cells were studied: one set of cells were located in the dark, while the second set of cells were irradiated at 530 nm. As predicted, *cis*-configuration recovered from the second set. Additionally, substantial degradation of BRD2 was recorded with *cis*-configuration of photoPROTAC-1-treated cells under 415 nm, indicating that incorporating *o*-F$_4$-azobenzenes as photoswitches in the PROTAC linker region was successful in providing the reversible spatiotemporal control over protein degradation.

### 4.3. Photoswitchable PROTACs in Immunophilins

To enhance the spectrum of optical control to targeted protein degradation, Trauner's and Pagano's research groups utilized photoswitchable PROTACs for prolyl *cis–trans* isomerase (FKBP12 protein [70]) [50]. They synthesized six structures (PHOTAC-II-1–6, as shown in Figure 14), where a lenalidomide-based structure was used as an E3 ligase ligand (in the case of five photoswitchable PROTACs **PHOTAC-II-1–5**), while a pomalidomide-based structure was used as an E3 ligase ligand in the chemical design of **PHOTAC-II-1–6**. The authors kept POI inhibitor as **SLF** (a synthetic ligand of FKBP12) common for **PHOTAC-II-1–6**. However, the azobenzene photoswitch varied in its positioning, wherein it was tethered adjacent to the E3 ligase ligand in the case of **PHOTAC-II-1–5**, whereas it was placed in the middle of **SLF** and the E3 ligase ligand in the case of **PHOTAC-II-6**. On the basis of the biological evaluation, **PHOTAC-II-5** and **PHOTAC-II-6** showed effective activity among six photoswitchable PHOTACs. Upon pulse irradiation (λ = 390 nm) by **PHOTAC-II-5**, significant cellular FKBP12 protein levels were reduced, but the same observation was not found in the dark condition. A general observation of attaining comparatively higher protein degradation activity for **PHOTAC-I-1–8** (for bromodomain proteins) and **PHOTAC-II-5** (for FKBP12 protein) than other photoswitchable PROTACs, and on the basis of their structural resemblance, the role of the positioning of azoswitch tethering to the E3 ligase ligand could be directed. Conversely, with **PHOTAC-II-6**, the azobenzene photoswitch positioned in the middle of the linker region showed a pattern similar to **PHOTAC-II-5** of strong light-dependent FKBP12 protein degradation and a mild activity in the dark (at 24 h), suggesting (*E*)-isomer presence in the solution. The competitive assay of **PHOTAC-II-5** and **PHOTAC-II-6** with **MLN4924** showed their inability to degrade the FKBP12 protein, suggesting the mechanism of CRBN mediation. Increased concentration of the PROTACs leads to a pronounced *hook effect* during experiments.

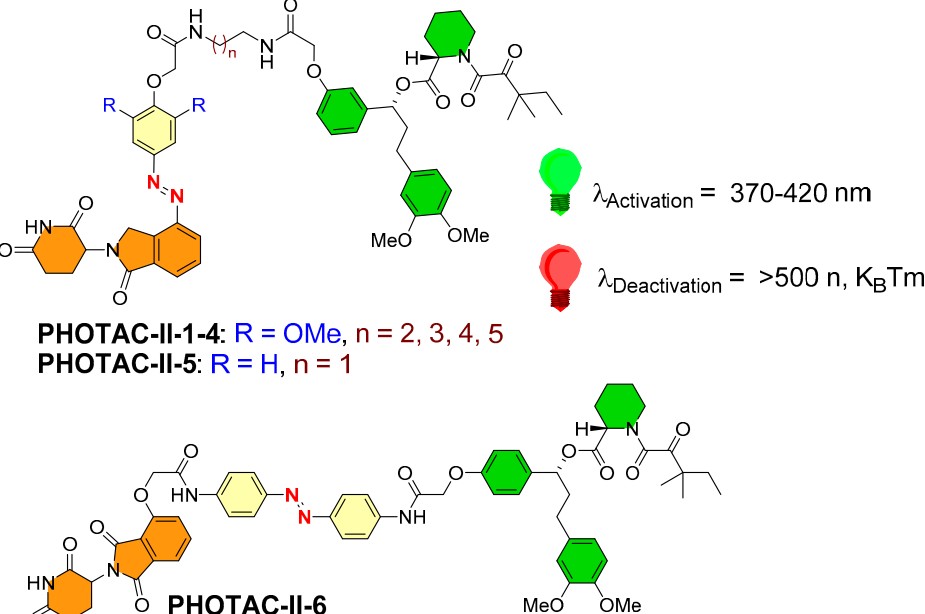

**Figure 14.** Chemical design of photoswitchable PROTACs for FKBP12 protein.

## 5. Summary and Outlook

Electromagnetic radiation has always been of keen interest for chemists in the field of imaging and diagnostics. As higher frequency wavelength electromagnetic waves ionize the cells, their direct use in chronic diseases is not considered the first line of treatment. However, the UV–VIS radiations are optimized to have sufficient energy vectors to activate a bioactive molecule but not to disintegrate the chemical structure.

Typical drug design approaches commonly practice occupancy-driven pharmacology, where the inhibitor competitively binds to the protein of interest at a particular threshold concentration. Therefore, the success of conventional small-molecular inhibitor-based drugs highly depends on their sustained concentrations, which require continuous dosing. However, continuous dosing commonly increases the chance of off-target toxicities and could lead to drug resistance (primarily observed in chronic disorders, such as cancer and infections). Therefore, event-driven, pharmacology-based approaches were introduced to overcome the limitation of conventional small-molecule-based drug discovery. One of the most popular approaches is protein degradation by PROTAC. Rather than inhibiting the protein, PROTACs degrade the protein; thereby, no continuous dosing is required, and there is less proneness to the emergence of resistance. However, with such immense advantages, PROTACs are also flawed, with some limitations that prevent their direct clinical use. Systemic toxicity is the primary concern of PROTACs, where some concentration of PROTACs goes inside the normal cells and degrades the protein, thereby risking the severity of on-target toxicity in the normal cells.

Various photochemical strategies were developed as a part of photopharmacology, but photocaged and photoswitches are the most successful ones. Photocaged PROTACs showed control of light-activating photolabile groups, which can be selectively activated with a specific wavelength of light. Deiter's research group used a coumarin-based photocaged VHL ligand to produce ERRα **PROTAC-2** to target estrogen-related receptor α (ERRα), which released at ($\lambda \leq 405$ nm), showing anticancer activity in breast cancer MCF-7 cell lines. Later, the same research group implemented another PPG nitropiperonyloxymethyl (NPOM) as CRBN-based BRD4 **PROTAC4**, which becomes activated at $\lambda = 365$ nm and degrades BRD4 bromodomain protein in HEK293T and 22Rv1 cells. Conversely, Xue et al. used a chemical design of dBET1 (a PROTAC) to transform it into photocaged PROTACs by incorporating 4,5-dimethoxy-2-nitrobenzyl (DMNB) as PPG. The resulting pc-PROTAC1 becomes activated upon irradiation ($\lambda = 365$ nm, 3 mW/cm$^2$). Further testing showed a broad cellular activity of pc-PROTAC1 in human Burkitt's lymphoma cells (Namalwa cells) and a liver cell line (HUH7 hepatocellular carcinoma cells). Xue et al. used a chemical template of MT-802 (a PROTAC) and incorporated it with 4,5-dimethoxy-2-nitrobenzyl (DMNB) as PPG to form **pc-PROTAC3** to find a photocaged-BTK PROTAC, which becomes activated at $\lambda = 365$ nm.

However, incorporating photolabile groups certainly provides spatiotemporal control, but systematic toxicity remains a concern, which led to the development of another strategy, namely, "photoswitchable-PROTACs". The presence of a photoswitch provides a reversible switchability between two photoisomeric states; therefore, the reversible active form can be converted back to inactive at a particular wavelength and vice versa. The choice of azobenzene was quite evident from previous reports in which azobenzene showed a fatigue resistance and comparatively smaller structural size that does not add to the molecular obesity of the whole molecular structure.

Some notable successes of photoswitches in PROTACs have been reported. Reynders et al. showed **PHOTAC-I-1-13** for bromodomain protein degradation [49] and PHOTAC-II-1-6 for immunophilins [49], while Jin et al. reported the **Azo-PROTAC-2C-6C** for kinase protein degradation [48]. The reversible photoswitching intrinsic property of photoswitchable PROTACs provides precise spatiotemporal control over targeted protein degradation and therefore significantly reduces the chance of systemic toxicities.

However, as both approaches use the PROTAC template where structures are generally larger in size, it potentially leads to typically higher molecular obesity (800–1200 Da).



Because of size, they tend to show limited cell permeability and tissue diffusion. However, choosing a cell-specific surface receptor (hybrid receptors, such as IGF-1/insulin receptor, HER) or intermembrane transporters [71] increases their cell delivery [72,73]. Furthermore, this provides them with an available biorthogonal chemistry handle that modifies their in-cell structure or conjugates with vitamin B coenzymes (especially pyridoxine and folic acid) or antibodies that enhance their cellular distribution and utilization [74]. The delivery route is essential in enhancing their cellular distribution, wherein the intraperitoneal and intravenous routes of administration showed reasonable tolerability [74]. Other formulations must be considered, such as a semiconducting polymer for nano-PROTAC activation in cancer therapy and nanoencapsulation by a biodegradable polymer [75,76].

UV light usually shows a limited tissue penetration, restricting reported light-activating PROTACs to skin disorders and other topical uses. Therefore, there is a growing interest in studying newer photoswitches and other possible light sources. Furthermore, in order to achieve optimal tissue penetration, red-shifted azobenzenes can use the near infrared region or the infrared region [77,78], and two-photon excitation processes could be incorporated for efficient photoswitching [68,79]. Recent advancements in implantable localized irradiation and optofluidic systems can certainly enhance the localized cellular activation of light-activating PROTACs, improving the scope of the other photomedicine strategies. Researchers also need to replace the azobenzene switch, which has poor in vivo application. However, azobenzenes have their characteristics that are based on their photoswitchability, which are mentioned in Section 4 (as "Photoswitches in PROTACs"). Other examples of photoswitches (fulgide, hydrazone, nobormadiene, spiropyrans, thioindigo, spirooxazine, diarylethene, merocyanine, etc.) and their conjugates can be used to replace the azobenzene photoswitch [80–83]. Azobenzenes are conventionally placed in the middle of the linker region (in between the PROTACs and E3 ligase), and therefore application of macrocyclization seems quite promising as it restricts the free rotation and thereby improves protein selectivity and physiochemical properties. Because of poor physiochemical properties of azobenzne structures, there is a tendency to show rapid in vivo hydrolysis, which also caused organic chemists to incorporate highly substituted aromatics [84] or a polyaromatic functional ring as one of the diazobenzene rings [85,86].

**Author Contributions:** Conceptualization, A.N.; writing—original draft preparation, A.N.; writing-review and editing, A.N., K.K.K. and A.S.V.-C.; funding acquisition, K.K.K. and A.N. All authors have read and agreed to the published version of the manuscript.

**Funding:** This research received no external funding.

**Institutional Review Board Statement:** Not applicable.

**Informed Consent Statement:** Not applicable.

**Data Availability Statement:** Not applicable.

**Conflicts of Interest:** The authors declare no conflict of interest.

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
