# Peer review of "Light-Activating PROTACs in Cancer: Chemical Design, Challenges, and Applications"

_applsci, doi:10.3390/app12199674_

Round 1
Reviewer 1 Report
In this review article, the authors summarize recent developments of light activating PROTAC molecules. The concepts of PROTAC, photocage-PROTAC and photoswitchable PROTAC are introduced in the manuscript (Figures 2 and 8). Several chemical structures of photocage-PROTACs (Figures 4–7) and photoswitchable PROTACs (Figures 9–14) are included in the manuscript. Bioactivities of some of these molecules to cells (cultured cells in most cases) are also described in the manuscript. The authors intend to use the light activating PROTACs in radiation therapies in cancer, although, in my impression, they do not sufficiently provide discussions about the feasibility and problems of this approach. It is interesting to design photoactive PROTAC derivatives to control their spatio-temporal activities, however, the current manuscript may not be adequately informative for researchers studying application of such molecules on cancer treatment. I also have major concerns about the strategy to apply the light activating PROTACs on cancer treatment. In my conclusion, this review article is not appropriate for publication in Applied Sciences, because it is rather oriented to basic research to control activities of PROTACs by light.
Major concerns
1) The authors claim that light activating PROTACs have advantages over conventional PROTACs by increasing selectivity to cancer cells. However, how to selectively illuminate cancer cells under in vivo conditions?
2) The authors also state that PROTACs are unlikely to have the emergence of resistance (line113). What is the rationale? I think resistant cells could appear if the target proteins have mutations that attenuate binding of the PROTAC molecule, although incidence of emerging resistant cells may depend on the type of the target proteins.
3) In the abstract, the authors state that ‘In this paper, we attempted to correlate the photophysical properties of phoswitchable-PROTACs (such as photoisomerization, quantum yield, and thermal relaxation states) to illustrate the photochemical-based structural activity relationships’. This sentence is not clear for me. In the main text, I only found explanations for these photophysical properties on some individual phoswitchable-PROTACs and did not get correlations among them. How it is useful in designing future phoswitchable-PROTAC molecules?
Minor concerns
4) line 82: part-A (yellow), blue?
5) line 84: part-C (blue), orange?
6) Figure 3 was not found in the manuscript.
7) line 338: Figure 5, Figure 7?
8) Figure 9 is not mentioned in the main text. It looks like a redundant figure.
9) Line 482, What is the meaning of “hook-effect”? Please include brief explanation in the manuscript.
10) Line 509, the word ‘PHOTAC’ have already appeared in line 372. The explanation of PHOTAC-I-13 in line 372 might be moved to later part of the manuscript.
11) Line 773, me?
Author Response
We thank the reviewer for taking the time and considering the paper. We address the
comments below.
The changes are marked in Yellow color in the manuscript.
In this review article, the authors summarize recent developments of light activating PROTAC molecules. The concepts of PROTAC, photocage-PROTAC and photoswitchable PROTAC are introduced in the manuscript (Figures 2 and 8). Several chemical structures of photocage-PROTACs (Figures 4–7) and photoswitchable PROTACs (Figures 9–14) are included in the manuscript. Bioactivities of some of these molecules to cells (cultured cells in most cases) are also described in the manuscript. The authors intend to use the light activating PROTACs in radiation therapies in cancer, although, in my impression, they do not sufficiently provide discussions about the feasibility and problems of this approach. It is interesting to design photoactive PROTAC derivatives to control their spatio-temporal activities, however, the current manuscript may not be adequately informative for researchers studying application of such molecules on cancer treatment. I also have major concerns about the strategy to apply the light activating PROTACs on cancer treatment. In my conclusion, this review article is not appropriate for publication in Applied Sciences, because it is rather oriented to basic research to control activities of PROTACs by light.
We understand the concern of the reviewer. First of all, we would like to clarify here that the paper is not based on radiotherapy or radiation for cancer treatment. However, “regarding feasibility and problems of this approach”, a section is dedicated and highlighted in cyan color (Page 4, from line 142 to line 154 )
Major concerns
- The authors claim that light activating PROTACs have advantages over conventional PROTACs by increasing selectivity to cancer cells. However, how to selectively illuminate cancer cells under in vivo conditions?
First of all, we are not claiming that light-activating PROTACs have illumination properties. However, light (radiation) is used as a source of activation for these photochemical PROTACs (or, in a simpler form, “ light activating PROTACs”). To explain further, the concept can be understood based on the Prodrug/drug concept analogy, where the prodrug remains inactive for the targeting protein/cell until it gets activated into the active form by using a specific wavelength of light.
PROTACs shortcomings are mentioned on Page 4 from lines 125 to 135 (marked in yellow color )
2) The authors also state that PROTACs are unlikely to have the emergence of resistance (line113). What is the rationale? I think resistant cells could appear if the target proteins have mutations that attenuate binding of the PROTAC molecule, although incidence of emerging resistant cells may depend on the type of the target proteins.
Additional information is added and revised accordingly (Text marked in yellow color)
3) In the abstract, the authors state that ‘In this paper, we attempted to correlate the photophysical properties of phoswitchable-PROTACs (such as photoisomerization, quantum yield, and thermal relaxation states) to illustrate the photochemical-based structural activity relationships’. This sentence is not clear for me. In the main text, I only found explanations for these photophysical properties on some individual phoswitchable-PROTACs and did not get correlations among them. How it is useful in designing future phoswitchable-PROTAC molecules?
The sentences are rewritten in the abstract section, and we also tried our best to simplify these sections in the running manuscript where these terminologies are mentioned.
Minor concerns
4) line 82: part-A (yellow), blue?
Revised accordingly (marked in yellow color)
5) line 84: part-C (blue), orange?
Revised accordingly (marked in yellow color)
6) Figure 3 was not found in the manuscript.
Figure 3 is now added to the manuscript
7) line 338: Figure 5, Figure 7?
The line was trivial as now removed.
8) Figure 9 is not mentioned in the main text. It looks like a redundant figure.
Revised accordingly (marked in yellow color)
9) Line 482, What is the meaning of “hook-effect”? Please include brief explanation in the manuscript.
A paragraph is added and marked in yellow color
10) Line 509, the word ‘PHOTAC’ have already appeared in line 372. The explanation of PHOTAC-I-13 in line 372 might be moved to later part of the manuscript.
The line is modified
11) Line 773, me?
Replaced the word “me” with “them”

Reviewer 2 Report
Negi et al., submitted manuscript "Light activating PROTACs in cancer: Chemical design, challenges, and applications" is about the photochemical activation of PROTACs, where authors considered two photo pharmacological approaches, for example: photoswitchable and photocagingThe strength of the paper is compiling the available literature based on photopharmacology-based PROTACs and their application to cancer cells.
Minor comments
1. In Figure 2, Photocaged PROTACs should "B" while photoswitchable PROTACs should be "C."
2. There is no description of Figure 3 in the main body of the paper and missing in the manuscript.
3. Molecular modeling figures (Figure 4A, 5A ) were shown in photocaging PROTAC; please specify the adaption source correctly in the figure caption.
4. Figure 1 seems blurred; please improve the quality.
5. There are complex sentences in the paper; use simple sentences to improve the paper's readability.
6. As PROTAC strategies are based on event-driven pharmacology, which authors showcased nicely in the paper. However, there are some limitations these event-driven pharmacology-based approaches cannot be applied, and only occupancy-driven pharmacology (conventional small molecular inhibitors) are helpful. For example, PROTAC strategies highly depend on the intracellular ubiquitin system, while some classical targets (such as GPCR) don't have any ubiquitination system in their proximity. Therefore, please include a paragraph showing the limitation of PROTACs as event-driven pharmacology where conventional approaches (occupancy-driven pharmacology) are applicable.
7. The success of photoswitches is limited because of the availability of diazo benzene groups as photoswitches, which are sensitive to physiological hydrolysis. Therefore, the authors need to add a paragraph to comment on the stability of diazobenzene photoswitches.
Author Response
We thank the reviewer for taking the time and considering the paper. We address the
comments below.
The changes are marked in Cyan color in the manuscript.
Negi et al., submitted manuscript "Light activating PROTACs in cancer: Chemical design, challenges, and applications" is about the photochemical activation of PROTACs, where authors considered two photo pharmacological approaches, for example: photoswitchable and photocaging
The strength of the paper is compiling the available literature based on photopharmacology-based PROTACs and their application to cancer cells.
Minor comments
- In Figure 2, Photocaged PROTACs should "B" while photoswitchable PROTACs should be "C."
Revised accordingly, marked in cyan color.
- There is no description of Figure 3 in the main body of the paper and missing in the manuscript.
Added into the manuscript main body as well as its illustration as Figure 3 (marked in cyan color)
- Molecular modeling figures (Figure 4A, 5A ) were shown in photocaging PROTAC; please specify the adaption source correctly in the figure caption.
Revised accordingly
- Figure 1 seems blurred; please improve the quality.
Figure 1 is now sketched by authors.
- There are complex sentences in the paper; use simple sentences to improve the paper's readability.
We went through and revised the sections to improve the readability of the paper.
- As PROTAC strategies are based on event-driven pharmacology, which authors showcased nicely in the paper. However, there are some limitations these event-driven pharmacology-based approaches cannot be applied, and only occupancy-driven pharmacology (conventional small molecular inhibitors) are helpful. For example, PROTAC strategies highly depend on the intracellular ubiquitin system, while some classical targets (such as GPCR) don't have any ubiquitination system in their proximity. Therefore, please include a paragraph showing the limitation of PROTACs as event-driven pharmacology where conventional approaches (occupancy-driven pharmacology) are applicable.
Section 3 is now presented and marked in cyan, which describes the common issues especially delivery with PROTAC strategy
- The success of photoswitches is limited because of the availability of diazo benzene groups as photoswitches, which are sensitive to physiological hydrolysis. Therefore, the authors need to add a paragraph to comment on the stability of diazobenzene photoswitches.
A paragraph is added in the conclusion section marked in cyan color

Round 2
Reviewer 1 Report
The authors have addressed the reviewer’s original concerns with minor changes in the revised manuscript. However, the changes do not make me change my conclusion. See below for my impression on the author’s responses and changed points.
We understand the concern of the reviewer. First of all, we would like to clarify here that the paper is not based on radiotherapy or radiation for cancer treatment. However, “regarding €i0feasibility and problems of this approach€i0”, a section is dedicated and highlighted in cyan color (Page 4, from line 142 to line 154 )
The authors state that the paper is not based on radiotherapy or radiation for cancer treatment. I agree with that. I also think that the light-activating PROTACs are interesting approach and useful as chemical biology tools.
However, the abstract of the paper starts from the following sentences implying the therapy applications, “Nonselective cell damage remains a significant limitation of radiation therapies in cancer. Technological advancement in radiation therapies for biomedical use led to the development of photopharmcology-based approaches targeting cancer cells selectively.” When I read the manuscript, I misunderstood that these sentences suggested that the strategy introduced in this paper (light-activating PROTACs) had a potential to improve the cancer cell selectivity in the treatment of the disease. Such misunderstanding resulted in my original concerns.
Even after considering this misunderstanding, I still conclude that this paper is not appropriate for publication in Applied Sciences Journal because it is oriented to basic research. Before submitting the manuscript to other Journals, I recommend to revising the abstract to avoid such a misleading on this approach. Related to this point, the title is also misleading and should be revised. The results introduced in this paper is about “cancer cells” but not about “cancer disease”. Just “cancer” in the current title may be confusing for the readers.
The authors adequately responded to the Major concerns 1) and 2), and the Minor concerns.
I still have a question on the Major concern 3).
3) In the abstract, the authors state that ‘In this paper, we attempted to correlate the photophysical properties of phoswitchable-PROTACs (such as photoisomerization, quantum yield, and thermal relaxation states) to illustrate the photochemical-based structural activity relationships’. This sentence is not clear for me. In the main text, I only found explanations for these photophysical properties on some individual phoswitchable- PROTACs and did not get correlations among them. How it is useful in designing future phoswitchable-PROTAC molecules?
The sentences are rewritten in the abstract section, and we also tried our best to simplify these sections in the running manuscript where these terminologies are mentioned.
I found the revised sentence in line24-27 in the abstract. In my impression, the original sentence should be deleted to complete this revision ‘In this paper, we attempted to correlate the photophysical properties of photoswitchable-PROTACs (such as photoisomerization, quantum yield, and thermal relaxation states) to illustrate the photochemical-based structural activity relationships’.
Author Response
The authors have addressed the reviewer’s original concerns with minor changes in the revised manuscript. However, the changes do not make me change my conclusion. See below for my impression on the author’s responses and changed points.
We understand the concern of the reviewer. First of all, we would like to clarify here that the paper is not based on radiotherapy or radiation for cancer treatment. However, “regarding €i0feasibility and problems of this approach€i0”, a section is dedicated and highlighted in cyan color (Page 4, from line 142 to line 154 )
The authors state that the paper is not based on radiotherapy or radiation for cancer treatment. I agree with that. I also think that the light-activating PROTACs are interesting approach and useful as chemical biology tools.
However, the abstract of the paper starts from the following sentences implying the therapy applications, “Nonselective cell damage remains a significant limitation of radiation therapies in cancer. Technological advancement in radiation therapies for biomedical use led to the development of photopharmcology-based approaches targeting cancer cells selectively.” When I read the manuscript, I misunderstood that these sentences suggested that the strategy introduced in this paper (light-activating PROTACs) had a potential to improve the cancer cell selectivity in the treatment of the disease. Such misunderstanding resulted in my original concerns.
Even after considering this misunderstanding, I still conclude that this paper is not appropriate for publication in Applied Sciences Journal because it is oriented to basic research. Before submitting the manuscript to other Journals, I recommend to revising the abstract to avoid such a misleading on this approach. Related to this point, the title is also misleading and should be revised. The results introduced in this paper is about “cancer cells” but not about “cancer disease”. Just “cancer” in the current title may be confusing for the readers.
We combine several methods where we compiled the studies from UV, IR irradiations to activate the biological activity of the molecules (PROTACs). Furthermore, these PROTACs are of two types based on their activation-specific type (Photocaged PROTACs, and Photoswitchable PROTACs). Photocaged PROTACs contain a photolabile group which get cleaved upon irradiation of specific wavelength, thereby providing a spatiotemporal control over cancer specific proteins. While photoswitchable PROTACs are one step ahead which provide more precise spatiotemporal control over degradation of cancer proteins in reversible manner (as it can allow different photophysical properties at different wavelength). Both of these types of PROTACs are explained to best of our understanding in the current manuscript, therefore, clearly encompass relevant applied fields of research.
We highlighted the cancer cell lines derived diseases (breast cancer, prostate cancer, liver cancer, blood cancer, others) used in these studies throughout in the current manuscript, which are highlighted in “red color”
The authors adequately responded to the Major concerns 1) and 2), and the Minor concerns.
I still have a question on the Major concern 3).
3) In the abstract, the authors state that ‘In this paper, we attempted to correlate the photophysical properties of phoswitchable-PROTACs (such as photoisomerization, quantum yield, and thermal relaxation states) to illustrate the photochemical-based structural activity relationships’. This sentence is not clear for me. In the main text, I only found explanations for these photophysical properties on some individual phoswitchable- PROTACs and did not get correlations among them. How it is useful in designing future phoswitchable-PROTAC molecules?
The sentences are rewritten in the abstract section, and we also tried our best to simplify these sections in the running manuscript where these terminologies are mentioned.
I found the revised sentence in line24-27 in the abstract. In my impression, the original sentence should be deleted to complete this revision ‘In this paper, we attempted to correlate the photophysical properties of photoswitchable-PROTACs (such as photoisomerization, quantum yield, and thermal relaxation states) to illustrate the photochemical-based structural activity relationships’.
This sentence is revised, and these words are deleted, now.

Round 3
Reviewer 1 Report
Both of these types of PROTACs are explained to best of our understanding in the current manuscript, therefore, clearly encompass relevant applied fields of research.
I agree that the manuscript describes the photoactivable PROTAC methods very well and the approach is scientifically interesting. However, It is not clear for me what kind of applications is possible by using this approach, even after the revisions. I think explanation of existing studies does not necessarily verify their utilities in the future. In my conclusion, this review should be submitted to other journals related to chemical biology field after revision of the title, which appears to be misleading. This paper introduces activities of photoactivable PROTAC compounds to cancer cells, however, it is not based on radiotherapy or radiation for cancer therapy. Cancer cells are actually related to cancer disease, however, I do not find rationale for improving the cancer cell selectivity in the cancer therapy by using the photoactivable PROTAC approach.